# Spatial mapping of immune cell environments in *NF2*-related schwannomatosis vestibular schwannoma

Adam P. Jones[1,2,8], Michael J. Haley[1,2,8], Miriam H. Meadows [1,2],
Grace E. Gregory[2,3], Cathal J. Hannan[2,4], Ana K. Simmons [1], Leoma D. Bere[1,2],
Daniel G. Lewis[2,4], Pedro Oliveira[5], Miriam J. Smith [2,6], Andrew T. King [2,4],
D. Gareth R. Evans[2,6], Pawel Paszek [1,7], David Brough [2,3,9] ✉,
Omar N. Pathmanaban [2,3,4,9] ✉ & Kevin N. Couper [1,2,9] ✉

*NF2*-related Schwannomatosis (*NF2* SWN) is a rare disease characterised by the growth of multiple nervous system neoplasms, including bilateral vestibular schwannoma (VS). VS tumours are characterised by extensive leucocyte infiltration. However, the immunological landscape in VS and the spatial determinants within the tumour microenvironment that shape the trajectory of disease are presently unknown. In this study, to elucidate the complex immunological networks across VS, we performed imaging mass cytometry (IMC) on clinically annotated VS samples from *NF2* SWN patients. We reveal the heterogeneity in neoplastic cell, myeloid cell and T cell populations that co-exist within VS, and that distinct myeloid cell and Schwann cell populations reside within varied spatial contexts across characteristic Antoni A and B histomorphic niches. Interestingly, T-cell populations co-localise with tumour-associated macrophages (TAMs) in Antoni A regions, seemingly limiting their ability to interact with tumorigenic Schwann cells. This spatial landscape is altered in Antoni B regions, where T-cell populations appear to interact with PD-L1+ Schwann cells. We also demonstrate that prior bevacizumab treatment (VEGF-A antagonist) preferentially reduces alternatively activated-like TAMs, whilst enhancing CD44 expression, in bevacizumab-treated tumours. Together, we describe niche-dependent modes of T-cell regulation in *NF2* SWN VS, indicating the potential for microenvironment-altering therapies for VS.

Bilateral vestibular schwannomas (VS) are the pathognomonic hallmark of the tumour predisposition syndrome *NF2*-related Schwannomatosis (*NF2* SWN)[1]. *NF2* SWN is caused by germline or tissue mosaic pathogenic variants (PV) in the tumour suppressor gene *NF2*, encoding a Moesin-Ezrin-Radixin-like protein (termed Merlin), which results in development of bilateral VS in >95% of afflicted patients[2–4]. The severity of the phenotype of *NF2* SWN[5] depends, in part, on the type of PV in the *NF2* gene, with whole gene deletions and missense variants resulting in a milder phenotype (later onset of symptoms and lower tumour burden), compared with truncating and frameshift variants, which cause the severe phenotype characterised by early onset of symptoms and high tumour burden[6–9].

Whilst VS are benign intracranial tumours, their development and treatment (surgical and radiation[10,11]) often cause severe morbidity

A full list of affiliations appears at the end of the paper. ✉e-mail: David.brough@manchester.ac.uk; omar.pathmanaban@manchester.ac.uk; kevin.couper@manchester.ac.uk

such as sensorineural hearing loss and facial weakness, and left unchecked, may lead to brainstem compression and hydrocephalus, requiring urgent neurosurgical intervention[12–14]. Currently, there are no approved drugs for the treatment of *NF2* SWN or VS. Multiple studies have trialled the use of bevacizumab, an inhibitor of angiogenesis, in *NF2* SWN-related VS and found a reduction in tumour growth and hearing stabilisation or improvement[15–17]. However, post-treatment toxicities such as hypertension, proteinuria and renal impairment are common, and many individuals treated with bevacizumab become refractory, resulting in uncontrolled growth and progressive hearing loss[18–20]. The current lack of alternative treatments for *NF2* SWN-related VS is a consequence of our limited understanding of the biology of *NF2* SWN-related VS tumours.

Whilst the tumour architecture of VS has been defined at the pathological level with the discovery of hypercellular Antoni A regions, and less cellular, loosely organised Antoni B regions[21], the cellular landscape in Antoni A and B regions, in particular the compartmentalisation and activities of immune cells within these distinct regions, has yet to be investigated. Notably, VS tumours harbour significant numbers of macrophages and T cells[22,23]. It has been suggested that macrophages in VS have a suppressive tumour-associated macrophage (TAM) phenotype[22] and the abundance of TAMs has been linked with VS tumour growth rate, potentially through vascular endothelial growth factor (VEGF) expression and promotion of angiogenesis[22,24,25]. A recent transcriptomic study identified signatures associated with immune enrichment and CD8+ T-cell senescence in rapidly progressing VS[26]. Despite these observations, there is a paucity of data describing how macrophages, T cells and neoplastic Schwann cells compartmentalise and interact within the VS tumour microenvironment (TME) and histomorphic niches.

In this study, we have employed Hyperion imaging mass cytometry (IMC) to spatially map the immunological landscape in Antoni A and B regions of *NF2* SWN VS tumours. We found that the spatial interactome (i.e. the network of cellular interactions defined by proximity) of Antoni A and B regions differ, whereby the perivascular niche in Antoni B regions is disrupted and loses connectivity to supportive immune cells, suggesting potential vascular degeneration in these regions. T cells also have distinct spatial profiles in Antoni A and B regions, highlighting two niche-dependent regulatory networks in which T cells exist within VS. Notably, we identified that disease severity and *NF2* pathogenic variants did not appear to have any significant influence on the TME, whilst bevacizumab-treatment was associated with a significant increase in CD44+ Schwann cells, suggesting a potential increase in matrix remodelling within these tumours. Collectively, our results provide insight into the immunological configuration of *NF2* SWN VS tumours and suggest tumour region-specific pathways that are potential therapeutic targets for the disease.

## Results

### Imaging mass cytometry highlights the intra and inter-heterogeneity across histomorphic niches in *NF2* SWN-related vestibular schwannoma

To explore the single-cell and spatial heterogeneity of VS in *NF2* SWN, 13 treatment naïve VS cases from individuals with *NF2* SWN (Table 1) were analysed within a programme of work outlined in Fig. 1A. H&E tissue sections were independently evaluated and validated by two neuropathologists, and annotated as either Antoni A or B regions (Fig. 1B), and incorporated into a 86-core TMA, with 2-8 cores per case. Antoni B regions were rarer than Antoni A regions in our TMA samples and in total, 63 Antoni A regions (from 12 different cases) were analysed and 10 Antoni B regions (from 6 different cases) were analysed (13 mixed Antoni A/Antoni B transition zone ROIs were subsequently omitted for direct comparisons between Antoni A and Antoni B regions). Sections were stained with a panel of 40 metal-conjugated

antibodies (Table 2), which was designed to allow interrogation of Schwann cell, myeloid cell, lymphoid cell, and vascular-related populations. Representative images showing staining of Schwann cells (S100B), macrophages (Iba1), T cells (CD8), stroma (SMA) and proliferative cells (Ki-67) in Antoni A and B regions are shown in Fig. 1B.

Following single-cell segmentation using Ilastik, we validated the quality of our automated segmentation by comparing it to manual segmentation of the same cells using Jaccard Index (Supplementary Fig. 1A). Next, we analysed the expression of canonical markers on cells by UMAP. This showed that markers related to Schwann cell (S100B+, SOX-10+), myeloid (Iba1+, CD68+, CD163+), lymphoid (CD8+, CD4+), and vascular-related (SMA+, CD31+) populations were largely segregated into distinct clusters (Supplementary Fig. 1B). Using Leiden clustering (workflow outlined in Supplementary Fig. 1C) we identified 23 distinct populations across the 13 VS cases (Fig. 1C and Supplementary Fig. 1D). Of note, 7 Schwann cell populations, 7 myeloid populations, 4 lymphoid (T-cell) populations, and 2 vascular populations were identified, with the remainder classified as erythrocytes, proliferating cells, and other cells that could not be clearly resolved with the panel (other) (Fig. 1C, Source Data File, outlining the total number and percentage of cells, including unclassified other cells, across our dataset and in each case). Although the neutrophil marker CD66b was not in the IMC panel, CD11b^hi cells were classed as neutrophils, and (as CD11b can be expressed at lower levels by other cell types), this was confirmed by qualitative IHC analyses using CD11b and CD66b (Supplementary Fig. 2A). CD206+ Iba1+ myeloid cells were very rare in the *NF2* SWN VS cases and were exclusively associated with the vasculature (Supplementary Fig. 2B). Consequently, CD206 expression was averaged lowly in identified myeloid populations (Fig. 1C) and the rare CD206+ Iba1+ cells could not be separately sub clustered at reasonable resolution. In general, Schwann cells appeared to express lower GLUT1 and MCT4, measures of glucose utilisation and lactate transport, than tumour associated macrophage and T cell populations (Fig. 1C). Whilst the signalling molecule pERK was lowly expressed on most cell populations, except a subset of Schwann and macrophage populations, the majority of pERK+ cells also expressed CAIX, indicative of response to hypoxia in these cells (Fig. 1C). Indeed, CAIX may be a more accurate measure of hypoxic response in these cells than Hif1α expression (which was largely negative in the dataset, apart from on CD11b^hi classified neutrophils), as has previously been shown[27]. Notably, the regulatory molecule Tim3 was expressed by multiple cell populations, indicative of its broad immune activities, and the pro-tumourogenic molecule TCIRG1, which has been correlated with immune infiltration within other tumour-types[28], was also highly expressed by various cell populations (Fig. 1C).

The cell clustering and cell signatures suggested that myeloid cells and Schwann cells exist on a spectrum of activation and polarisation states in VS, indicating that the myeloid cells exhibit both pro-inflammatory and regulatory phenotypes. As such, we classified two hybrid macrophage populations, which could not be effectively subclustered at higher resolution into discrete and robust subpopulations using the markers within our IMC panel, as classically activated-like cells (Iba1+, HLA-DR+, CD74+, CD163−) and alternatively activated-like cells (Iba1+, CD68+, HLA-DR+, CD163+, Tim3+), rather than as polarised definitive populations (Fig. 1C).

Given that Schwann cells and alternatively activated-like macrophages have atypical morphologies and can be intertwined with each other and with other cell types (including T cells) in the VS tissue (Fig. 1B), and as certain molecules (such as S100B) can be secreted locally within tissue niches, it was very challenging to completely segregate respective marker expression for alternatively activated-like macrophages (characterised primarily through IBA-1, CD163 expression) and Schwann cells (Iba-1−, CD68−, HLA-DR−, S100B+, SOX10+/−) during segmentation, and as such we witnessed some S100B crossover onto alternatively activated-like TAMs (Fig. 1C). We also observed

**Table 1 | Case information**

| Case | Sex | Age at radiological diagnosis | Age at surgery | NF2 pathogenic variant (NM_000268.4) | Disease severity | Prior treatments | bevacizumab treatment | | | |
|---|---|---|---|---|---|---|---|---|---|---|
| | | | | | | | Dose | Regime | Duration | Cessation prior to surgery |
| NF2VS1 | M | 18-24 | 25-34 | Unknown (likely mosaic) | Mild | None | | | | |
| NF2VS2 | M | 55-64 | 55-64 | c.999±1G>T, p.(?) | Mild | None | | | | |
| NF2VS3 | F | 45-54 | 45-54 | Whole gene deletion | Mild | Resection | | | | |
| NF2VS4 | F | 25-34 | 35-44 | c.1604T>C, p.(Leu535Pro) | Mild | None | | | | |
| NF2VS5 | F | 35-44 | 45-54 | c.1229_1241del, p.(Gln410Profs*12) | Severe | None | | | | |
| NF2VS6 | M | 45-54 | 45-54 | c.516±1G>T, p.(Leu361Cysfs*3) | Moderate | None | | | | |
| NF2VS7 | M | 18-24 | 18-24 | Deletion of promotor to exon 1 | Mild | None | | | | |
| NF2VS8 | F | 25-34 | 25-34 | c.1080del, p.(Leu361Cysfs*3) | Severe | None | | | | |
| NF2VS9 | M | 25-34 | 25-34 | c.265G>T, p.(Glu89*) | Severe | None | | | | |
| NF2VS10 | M | 25-34 | 25-34 | c.1606C>T, p.(Gln536*) | Moderate | None | | | | |
| NF2VS11 | F | 25-34 | 25-34 | Deletion of exon 15 | Moderate | None | | | | |
| NF2VS12 | F | 25-34 | 25-34 | c.1021C>T, p.(Arg341*) | Severe | None | | | | |
| NF2VS13 | F | 18-24 | 25-34 | Deletion of intron 1 to exon 17 | Moderate | Resection | | | | |
| NF2VS14 | M | 10-17 | 10-17 | c.1396C>T, p.(Arg466*) | Severe | bevacizumab | 10 mg/kg | Every 2 weeks | 3-5 years | <10 weeks |
| NF2VS15 | M | 10-17 | 18-24 | Deletion of exons 7 to 14 | Moderate | bevacizumab | 10 mg/kg | Every 2 weeks | 3-5 years | 10-15 weeks |
| NF2VS16 | F | 18-24 | 25-34 | c.169C>T, p.(Arg57*) | Severe | bevacizumab | 7.5 mg/kg | Every 3 weeks | 3-5 years | 10-15 weeks |

some cross over between CD3, Iba1 and CD163 between myeloid cells and T cells (Fig. 1C).

To confirm the accuracy of the IMC-identified populations, we utilised MaxFuse[29] to integrate our IMC data with publicly available single-cell RNA sequencing data from NF2-mutated VS tumours[30] (Supplementary Fig. 3). From a total of 731,237 cells in our IMC dataset, we were able to align 659,880 cells (90.24%) (Fig. S3A) and IMC and single-cell RNA seq labels were successfully matched and mapped through MaxFuse for 548,455 cells (83.1%) (Supplementary Fig. 3B–D). This analysis provides orthogonal validation of the various cell populations characterised in our IMC dataset.

By IMC analyses, PD-L1$^+$ Schwann cells, proliferative PD-L1$^+$ Schwann cells, vascular endothelial cells, and hypoxic perivascular neutrophils were all found to be significantly more abundant in Antoni A regions compared to Antoni B regions (Fig. 1D). There was no significant difference in the remaining cell populations across the two histomorphic niches (Supplementary Fig. 4). Interestingly, we also observed that classically activated-like TAMs were significantly more abundant in both Antoni A and B regions compared to alternatively activated-like TAMs (Fig. 1E). Contrasting the relative proportions of the identified cell populations across the different case ROIs within our TMA revealed substantial intratumoral heterogeneity (with differences in abundance of cell populations in different Antoni A and B ROIs within the same tumour), as well as intertumoral heterogeneity (Fig. 1F), where Schwann cell and myeloid cell populations were particularly diverse across tumour regions and between cases. Given this, we addressed whether disease severity (NF2 pathogenic variants) contributed to the observed intertumoral heterogeneity. However, we observed no significant differences in relative cell abundances in Antoni A regions between severity groups, suggesting disease severity does not correlate with intertumoral heterogeneity (Supplementary Fig. 5).

**The interactions of myeloid cell populations with Schwann cell populations across histomorphic niches in NF2 SWN-related vestibular schwannoma**

Given that Schwann cells and macrophages are the predominant populations in VS (Fig. 1), and the apparent importance of macrophages in controlling VS growth rate[22,31], we next investigated how Schwann cell and myeloid cell populations compartmentalise and interact within the distinct Antoni A and B niches. Representative images of canonical Schwann cell (S100B, SOX-10), myeloid cell (Iba1, HLA-DR) and stromal markers (SMA), and the spatial positioning of identified Schwann cell, myeloid, and stromal populations within Antoni A and B regions are shown in Fig. 2A.

To quantitatively assess the identity of and extent of homotypic (between the same cell populations) and heterotypic (between different cell populations) cell–cell interactions between myeloid cell and Schwann cell populations across the histomorphic niches, we performed cross pair correlation functions (cross-PCF) analyses[32]. In Antoni A regions (Fig. 2B), myeloid and Schwann cell populations largely clustered with cells of the same type or with cells within the same sub-population group (i.e., myeloid cell populations with myeloid cells and Schwann cell populations with Schwann cells), and cells generally showed lower than expected (as if by chance) interactions with cell populations of different groups. In contrast, there were fewer statistically significant cell–cell interaction partners within Antoni B regions, with cell populations again showing preferential homotypic interactions with themselves (Fig. 2B). The significant interactions between hypoxic perivascular neutrophils, neutrophils, and classically activated-like TAMs was lost within the perivascular niche of Antoni B regions. Together, these data indicate there is substantial spatial variation in cell positioning and interactions between Antoni A and B regions. Whilst it

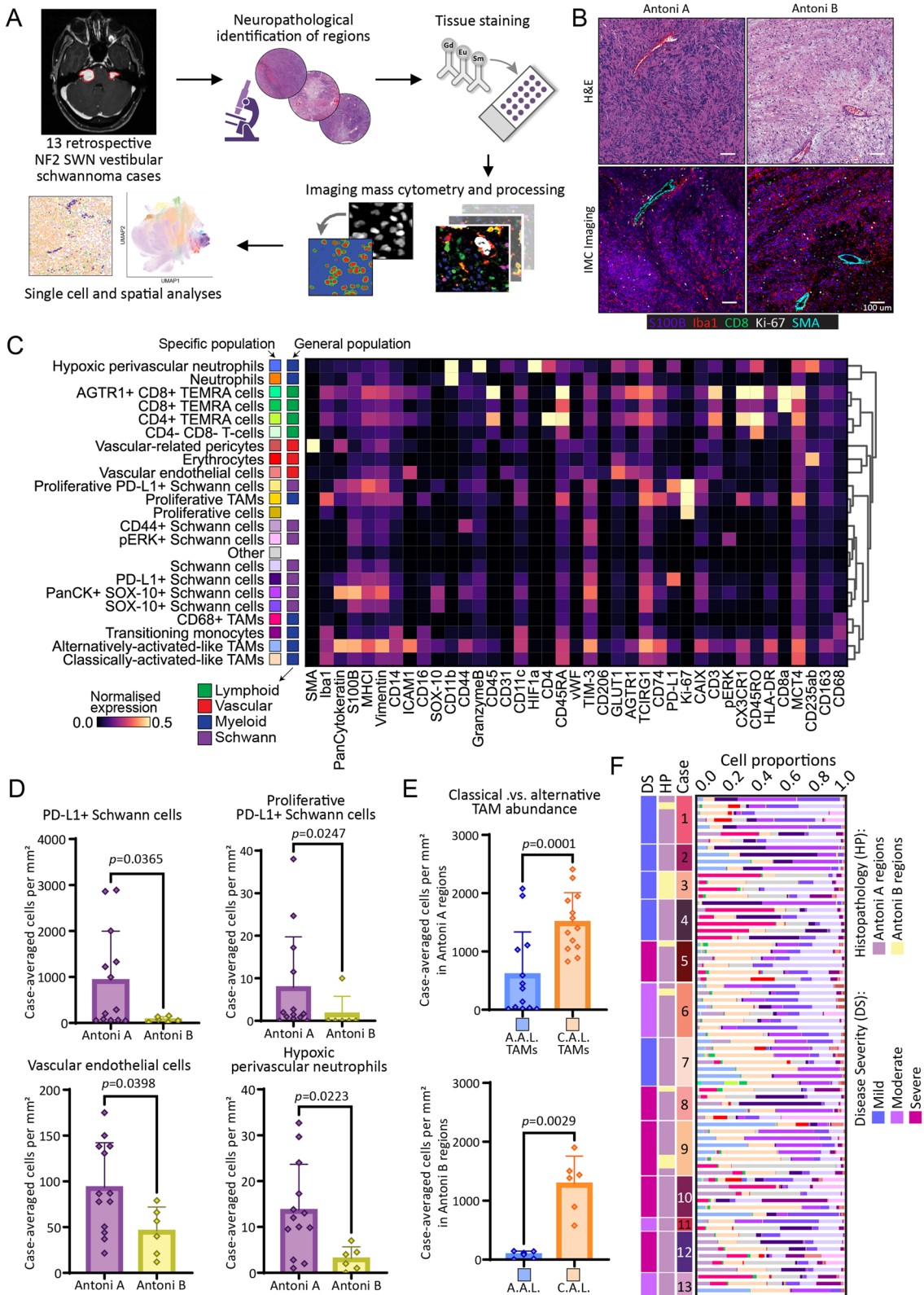

appears myeloid cells and Schwann cells have a preference to interact more with themselves across the histomorphic niches, there is more cell–cell interactivity and heterogeneity between myeloid cells and Schwann cells in Antoni A regions compared to Antoni B regions. The cellular landscape and cell–cell interactions within the perivascular niche also appear to differ across regions.

To expand upon the direct cell–cell interactivity analyses, we next investigated the cellular networks of myeloid cell populations with Schwann cell populations across Antoni A and B histomorphic niches, using adjacency cell network (ACN) analysis (Fig. 2C). As suggested from the cell–cell interaction analyses (Fig. 2B), classically activated-like TAMs and transitioning

**Fig. 1 | Imaging mass cytometry (IMC) workflow and generation of the single-cell atlas in NF2 SWN-related vestibular schwannoma. A** Schematic of IMC workflow for investigating *NF2* SWN-related VS: niche identification, tissue processing and staining, data acquisition and processing of raw data, single-cell segmentation, and downstream analyses. **B** Sequential haematoxylin and eosin (H&E) and IMC images of Antoni A and Antoni B regions, visualising the core Schwann cell (S100B; purple), macrophage (Iba1, red), T-cell (CD8α; green), vascular (SMA; cyan), and proliferation (Ki-67; white) markers. Scale bar representative of 100 μm. **C** Heatmap detailing expression pattern of all single-cell populations identified in IMC analysis via Leiden clustering. Scale bar indicating normalised expression (N.E.). **D** Comparison of case-averaged abundance of significantly different cell populations across Antoni A and Antoni B regions. The remaining non-significant population abundances by histopathology can be found in Fig. S4 ($n = 13$). **E** Comparison of alternatively activated-like TAMs versus classically activated-like TAMs across both Antoni A and Antoni B regions ($n = 13$). **F** Relative abundance of cell populations across ROIs, annotated by case, histopathology, and disease severity. SMA: smooth muscle actin. Statistical comparisons in D and E were made using two-tailed unpaired *t*-tests (for normally distributed data) or two-tailed Mann−Whitney *U* tests (result are the mean of the group + SD). Source data are provided as a Source Data file.

### Table 2 | IMC antibody panel

| Compartment | Acronym | Antigen | Target | Metal Tag | Dilution |
|---|---|---|---|---|---|
| Neoplastic | S100B | S100 calcium-binding protein B | Schwann cells | 141Pr | 1:200 |
| | SOX10 | SRY-related HMG-Box 10 | Schwann cells | 148Nd | 1:100 |
| | PanCK | Pan-cytokeratin | Tumour cells | 139La | 1:100 |
| Immune | | | | | |
| - Pan | CD45 | Protein tyrosine phosphatase receptor type C | Blood cells | 152Sm | 1:50 |
| | TIM-3 | T-cell immunoglobulin and mucin-domain containing-3 | Immunosuppression | 161Dy | 1:100 |
| | Granzyme B | – | Cytotoxicity | 151Eu | 1:100 |
| | CD16 | FcγRIII | Immune cells | 146Nd | 1:100 |
| | CX3CR1 | Fractalkine receptor | Immune cell migration | 172Yb | 1:100 |
| | CD11b | Integrin alpha M | Innate immune cells | 149Sm | 1:100 |
| | CD11c | Integrin alpha X | Immune cells | 154Sm | 1:50 |
| - Myeloid | CD14 | – | Monocytes/macrophages | 144Nd | 1:50 |
| | CD163 | – | Macrophages | 196Pt | 1:50 |
| | Iba1 | Ionised calcium-binding adaptor molecule 1 | Macrophages | 115Ln | 1:50 |
| | CD74 | HLA-II histocompatibility antigen gamma chain | Antigen-presenting cells | 166Er | 1:100 |
| | HLA-DR | Human leucocyte antigen DR | Antigen-presenting cells | 174Yb | 1:100 |
| | CD206 | Mannose receptor | Macrophages | 162Dy | 1:100 |
| | CD68 | – | Macrophages | 198Pt | 1:50 |
| - Lymphoid | CD3 | CD3 | T cells | 170Er | 1:50 |
| | CD4 | – | Helper T cells | 156Gd | 1:50 |
| | CD8α | CD8 alpha chain | Cytotoxic T cells | 175Lu | 1:100 |
| | CD45RA | – | Naïve/exhausted T cells | 158Gd | 1:100 |
| | CD45RO | – | Memory T cells | 173Yb | 1:100 |
| | TCIRG1 | T cell immune regulator 1 | T cells | 165Ho | 1:50 |
| Vasculature | SMA | Smooth muscle actin | Blood vessel structure | 89Y | 1:100 |
| | CD31 | Platelet endothelial cell adhesion molecule 1 | Blood vessel endothelium | 153Eu | 1:100 |
| | vWF | von Willebrand factor | Endothelial cell activation | 159Tb | 1:100 |
| Miscellaneous | HLA-A | Human leucocyte antigen A | Nucleated cells | 142Nd | 1:50 |
| | CD235ab | Glycophorin A/B | Erythrocytes | 195Pt | 1:50 |
| | Vimentin | – | Cell motility and stability | 143Nd | 1:100 |
| | CAIX | Carbonic anhydrase IX | Hypoxia | 169Tm | 1:100 |
| | HIF-1α | Hypoxia-inducible factor 1 alpha | Hypoxia | 155Gd | 1:50 |
| | GLUT1 | Glucose transporter 1 | Hypoxia and metabolism | 163Dy | 1:100 |
| | PD-L1 | Programme death-ligand 1 | Immune suppression | 167Er[a] | Neat[b] |
| | Ki-67 | – | Proliferation | 168Er | 1:100 |
| | AGTR1 | Angiotensin II type 1 receptor | Signalling | 164Dy | 1:100 |
| | pERK1/2 | Phosphorylated extracellular signal-regulated kinase 1/2 | Signalling | 171Yb | 1:100 |
| | MCT4 | Monocarboxylate transporter 4 | Signalling | 176Yb | 1:200 |
| | CD44 | Hyaluronan receptor | Signalling | 150Nd | 1:50 |

[a]PD-L1 was detected by a secondary anti-rabbit antibody conjugated with 167Er.
[b]100 μL of neat PD-L1 was used for staining. Specific antibody concentration is approximately 1.61μg/mL.

monocytes had the most heterogenous networks in Antoni A regions, with significant connectively with various Schwann cell and myeloid cell populations as well as vascular-related cells. These extensive networks were lost in Antoni B regions, where classically activated-like TAMs and transitioning monocytes only connected with each other and CD68⁺ TAMs, with transitioning monocytes also maintaining their relation to PD-L1⁺ Schwann cells. Interestingly, alternatively activated-like TAMs and CD68⁺

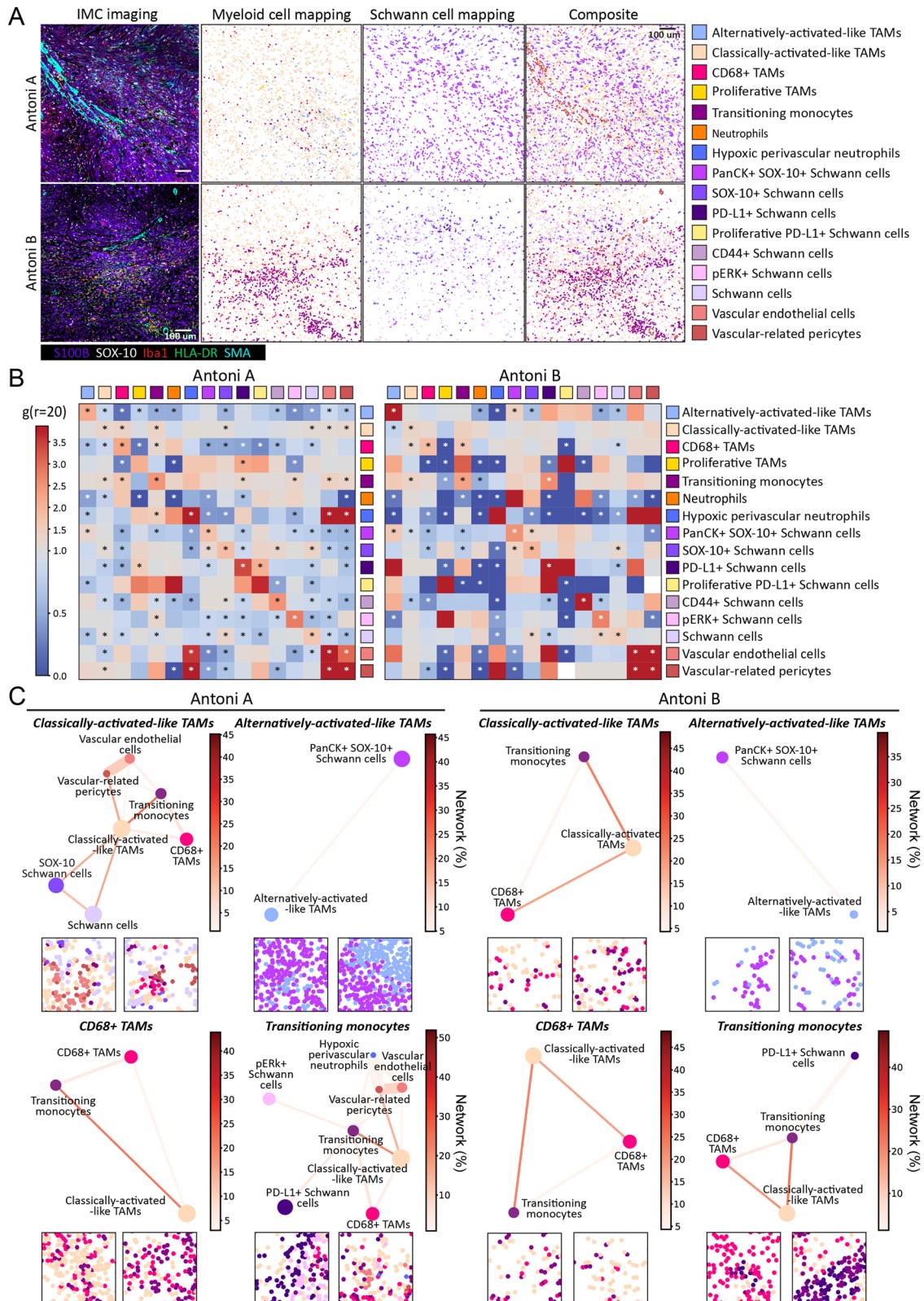

TAMs had the same networks in both Antoni A and B regions. Together, these analyses indicate distinct cellular networks that are either heterogenous (as with classically activated TAMs and transitioning monocytes) or homogenous (as with alternatively activated-like TAMs and CD68+ TAMs) between myeloid cells in Antoni A regions, which are lost (heterogeneous) or maintained (homogenous) in Antoni B regions.

**T-cell populations exist within distinct environments associated with infiltration and suppression across histomorphic niches in *NF2* SWN-related vestibular schwannoma**

T cells play a fundamental role in anti-tumoral immunity and tumour cell elimination[33], yet their role in VS is unclear. We identified different populations of CD8+ and CD4+ T cells within Antoni A and B regions (Fig. 1C), but what these populations interact with and if they occupy

**Fig. 2 | Spatial omics highlight the interactomes between Schwann cell and Myeloid cell populations in NF2 SWN-related vestibular schwannoma. A** From left to right, IMC visualisation of Schwann cells and myeloid cells across Antoni A and Antoni B regions. Markers used: S100B (purple), SOX-10 (white), Iba1 (red), HLA-DR (green), and SMA (cyan). Next, single-cell spatial seaborn maps of all Schwann cell and myeloid cell populations within Antoni A and B regions are shown, with a composite of the 2 major cell groups with the vascular networks. The scale bar indicates 100 μm. **B** Cross-pair correlation functions (PCF) heatmaps showing the Schwann cell and myeloid cell interaction; scale bar indicates strength of significant cell pair correlates. **C** Spatial connectivity plots indicating the significant networks of Schwann cell and myeloid cell populations, by adjacency cell network (ACN) analysis, across Antoni A and Antoni B histomorphic niches, with representative images of each network. Node size (coloured circle) indicates mean abundance for each cell cluster across all ROIs; lines connecting each node shows significant cell networks between cell types informed by ACN analysis. Line thickness associates the gr20 value to each significant cell pair, where the thicker the line, the higher the co-localisation of the cell populations. scale bar indicates strength of connectivity. **B, C** *$p < 0.05$, using cross-PCF and ACN statistical analyses, see Methods ($n = 13$). Source data are provided as a Source Data file.

the same tumour subniches is unknown. As such, we investigated the spatial interactions of T cells across the different histomorphic Antoni A and B niches in *NF2* SWN VS. Representative images of canonical T-cell (CD8, CD4), Schwann cell (S100B), macrophage (Iba1) and stromal (SMA) marker staining, and the spatial positioning of identified T-cell, Schwann cell, myeloid, and stromal populations within Antoni A and B regions are shown in Fig. 3A.

To statistically evaluate the direct cell−cell interactome of the four identified T-cell populations across histomorphic niches in VS tumours, we again employed cross-PCF analysis[32]. In Antoni A regions (Fig. 3B), the strongest interactions were seen between T-cell subtypes, with AGTR1+ CD8+ T effector memory re-expressing CD45RA (TEMRA) cells, CD8+ TEMRA cells, CD4+ TEMRA cells, and CD4-CD8- T cells all appearing to interact with themselves and each other. This indicates, as shown in Fig. 3A, that the different T-cell populations frequently intermix in Antoni A regions, rather than segregate within distinct areas. Interestingly, both CD8+ TEMRA populations interacted with vascular-related pericytes, whilst CD4+ TEMRA cells clustered with vascular endothelial cells. Both CD8+ TEMRA populations and CD4+ TEMRA cells had significant interactions with transitioning monocytes and classically activated-like TAMs, but only AGTR1+ CD8+ TEMRA cells significantly engaged with alternatively activated-like TAMs. Conversely, all T-cell populations except for AGTR1+ TEMRA cells significantly interacted with hypoxic perivascular neutrophils. As observed in Fig. 3A, the T-cell populations showed lower than expected interactions with all the different Schwann cell populations, suggesting T cells are predominantly sequestered within TAM-rich areas in Antoni A regions of the tumour. Interestingly, there were, in general, fewer clustering interactions across T-cell populations in Antoni B regions (Fig. 3B). AGTR1+ TEMRA cells and CD8+ TEMRA cells still exhibited significant homotypic and heterotypic interactions with themselves and each other, as well as with CD4+ TEMRA cells. On the other hand, CD4-CD8- T cells and CD4+ TEMRA cells exhibited lower statistical cell−cell interactions with other T-cell populations within Antoni B regions than observed in Antoni A regions. As opposed to the results in Antoni A regions, neither CD8+ TEMRA cells or CD4+ TEMRA cells interacted significantly with vascular-related pericytes or vascular endothelial cells in Antoni B regions, and only CD8+ TEMRA cells showed interactions with classically activated-like TAMs or transitioning monocytes. Instead, the CD8+ TEMRA cells and CD4+ TEMRA cells showed statistically enriched interactions with PD-L1+ Schwann cells within Antoni B regions. As PD-1 was not within the IMC panel, we verified through qualitative IHC analyses that CD8+ T cells express PD-1 within the VS tumours (Supplementary Fig. 6). Taken together, these results highlight the heterogeneity of T-cell interactions across different histomorphic niches in VS, particularly when addressing T-cell co-localisation with TAM populations and immunoregulatory PD-L1+ Schwann cells in Antoni A and B regions, respectively.

Given the apparent differences in cell−cell interactions and positioning of T-cell populations in Antoni A and B regions, we next investigated the cellular networks of T-cell populations across these histomorphic niches by ACN analysis (Fig. 3C). Across both histomorphic niches, T cells commonly had a strong network with each other, particularly in Antoni A regions. AGTR1+ CD8+ TEMRA cells had

connectivity with both classically activated-like and alternatively activated-like TAMs, transitioning monocytes and vascular-related cells in Antoni A regions, whereas in Antoni B regions, these T cells lost their network with vasculature and alternatively activated-like TAMs, but gained significant connectivity to PD-L1+ Schwann cells. CD8+ TEMRA cells had the most heterogeneous networks across both histomorphic niches, particularly in Antoni A regions, with connections to vascular-related cells, classically activated-like TAMs, transitioning monocytes, CD68+ TAMs and PD-L1+ Schwann cells. These networks were altered in Antoni B regions, where CD8+ TEMRA cells lost connectivity to the vascular network, most myeloid cells, and showed high connectivity with PD-L1+ Schwann cells. CD4+ TEMRA cells also lost their connectivity to vasculature (and transitioning monocytes) from Antoni A to Antoni B regions, and gained significant connections to PD-L1+ Schwann cells, whilst maintaining their connection to classically activated-like TAMs. Interestingly, the network between double negative CD4- CD8- T cells and other T-cell populations and vasculature in Antoni A regions was completely lost in Antoni B regions, with no significant network connections present. Together, these analyses suggest T cells in Antoni A regions exist within networks associated with infiltration and localisation with myeloid cells, whilst in Antoni B regions, these networks completely alter and T cells become significantly more associated with immunoregulatory PD-L1+ Schwann cells, highlighting distinct modes of T-cell regulation that are niche-dependent.

## CD8+ TEMRA cells and CD4+ TEMRA cells appear to be actively recruited and suppressed within *NF2* SWN-related vestibular schwannoma

Given the distinct spatial interactomes for T cell subsets that exist between Antoni A and B niches (Fig. 3), we next used CellPhone DB[34] to identify the nature of receptor-ligand interactions between both CD8+ TEMRA cells and CD4+ TEMRA cells and their respective identified co-localised cell partners. As the IMC dataset provided insufficient depth to power such analyses, we utilised the MaxFuse integrated scRNA-seq populations (Supplementary Fig. 3). We focused on CD8+ TEMRA and CD4+ TEMRA cells, given the accuracy of mapping between IMC and scRNA-seq populations and their likely important roles within the VS TME. We evaluated the receptor-ligand interactions - cell surface-bound and soluble−with a minimum percentage threshold of 5% across all CD8+ and CD4+ TEMRA cells.

The CD8+ TEMRA receptor-ligand map (Fig. 4A) showed multiple interactions involved in recruitment and infiltration of immune cells, through chemokine and integrin pathways. Within spatially confirmed cell−cell interactions within Antoni A regions (Fig. 3B), CCR5, CXCR4, and CCR6 on CD8+ TEMRA cells was predicted to interact with CCL3, CCL4, CCL5, CXCL12 and CCL20 expressed variably by classically activated-like TAMs, alternatively activated-like TAMs, transitioning monocytes, and CD4+ and CD8+ TEMRA cells (Fig. 4A). The predicted relationship between integrin expression (e.g. ITGA4, ITGA1) on CD8+ TEMRA cells and expression of extracellular matrix and adhesion molecules (e.g. VCAM-1, COL6A3 and COL15A1) on vascular-related pericytes also suggested the potential active recruitment of

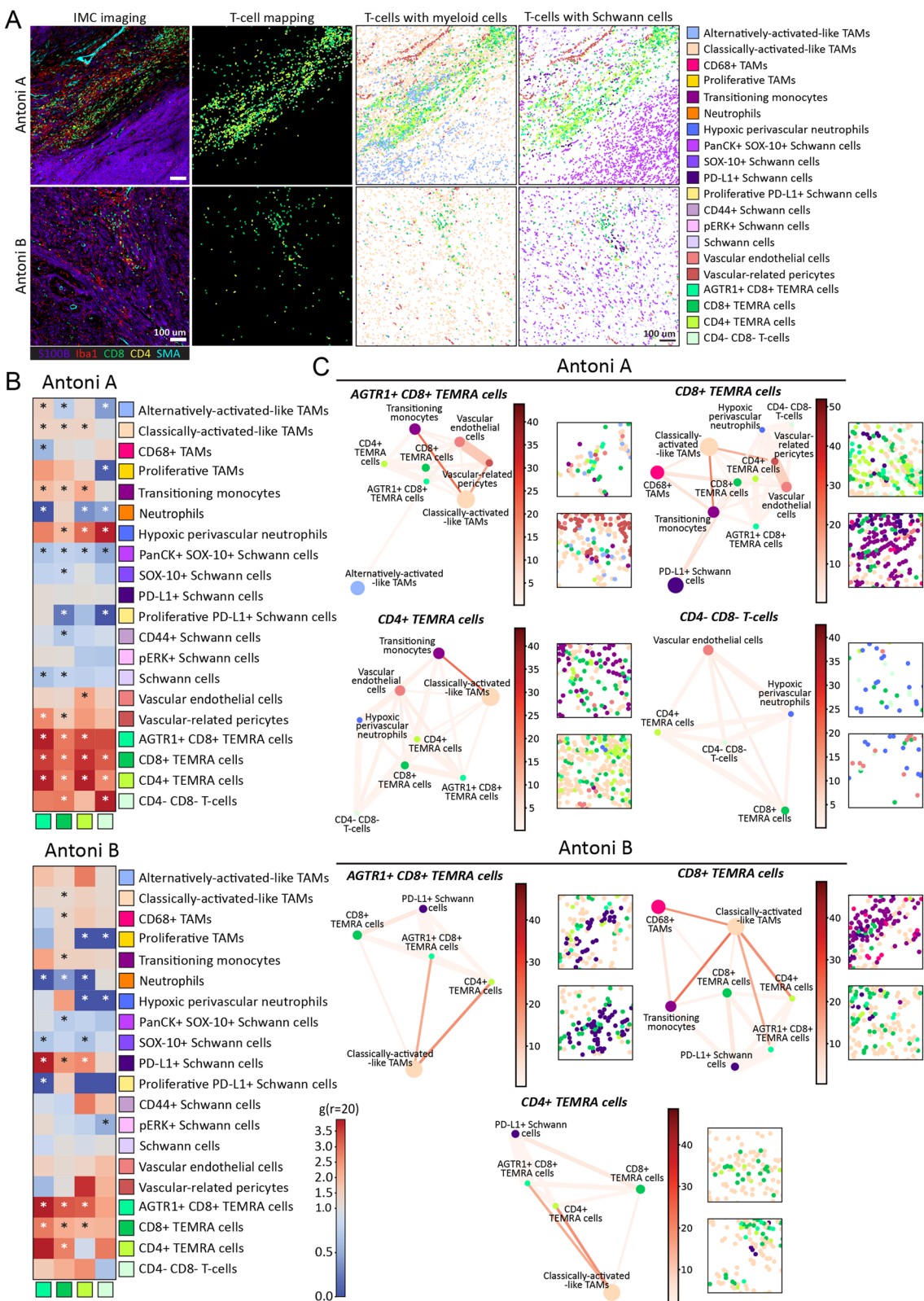

T cells through tumour vasculature in Antoni A regions. Within spatially confirmed cell–cell interactions within Antoni B regions (Fig. 3B), CD8[+] TEMRA cells and PD-L1[+] Schwann cells were predicted to interact through distinct receptor-ligand interactions, including L1CAM on PD-L1[+] Schwann cells (Fig. 4A), a neuronal adhesion molecule upregulated in cancer[35], which may be associated with CD8[+] TEMRA suppression in Antoni B regions.

Several T-cell inhibitory pathways were also identified within the receptor-ligand map for CD8[+] TEMRA cells, including CD52-SIGLEC10, SPP1-ITGA4, TGFB1-TGFBR1, and various KLR receptors including KLRB1, across myeloid, T-cell and Schwann cell populations, suggesting various routes of T-cell regulation across both Antoni A and B regions (Fig. 4A). Many of these pathways were similarly predicted for CD4[+] TEMRA cell communication with spatially-confirmed co-localised

**Fig. 3 | Spatial omics demonstrate the T-cell interactome across different histomorphic niches in NF2 SWN-related vestibular schwannoma. A** From left to right, IMC visualisation of T-cells across Antoni A and Antoni B regions. Markers used: S100B (purple), Iba1 (red), CD8α (green), CD4 (white) and SMA (cyan). Next, single-cell spatial seaborn maps of all T-cell populations, and co-localisation to Schwann cell or TAM populations with the vascular networks across Antoni A and B regions. Scale bar indicates 100 μm. **B** Cross-pair correlation functions (PCF) heatmaps showing T-cell interaction partners with all other cell populations; scale bar indicates strength of significant cell pair correlates. **C** Spatial connectivity plots

indicating the significant networks of T-cell populations, by adjacency cell network (ACN) analysis, across Antoni A and Antoni B histomorphic niches, with representative images of each network. Node size (coloured circle) indicates mean abundance for each cell cluster across all ROIs; lines connecting each node show significant cell networks between cell types informed by ACN analysis. Line thickness associates the gr20 value to each significant cell pair, where the thicker the line, the higher the co-localisation of the cell populations. the scale bar indicates strength of connectivity. **B**, **C** *$p < 0.05$, using cross-PCF and ACN statistical analyses, see Methods ($n = 13$). Source data are provided as a Source Data file.

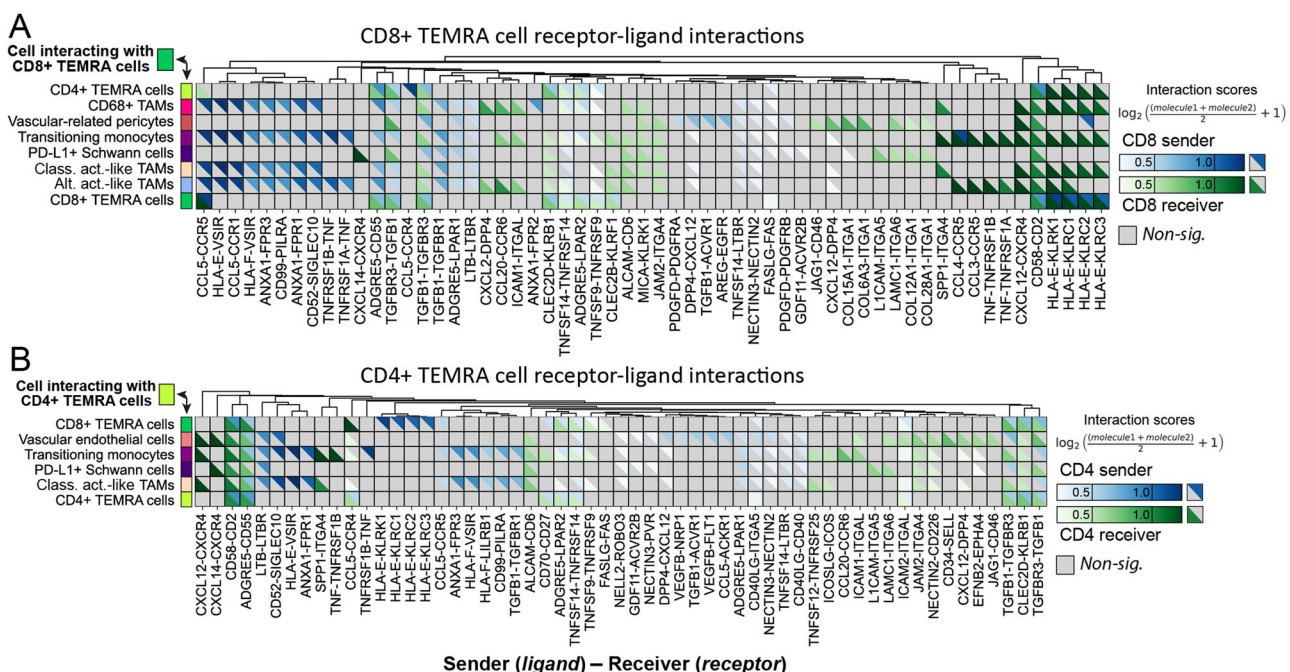

**Fig. 4 | Receptor-ligand interaction analyses for CD8+ and CD4+ TEMRA cell populations within VS.** Receptor-ligand analysis between **A** CD8[+] and **B** CD4[+] TEMRA cells and statistically confirmed co-localised cell populations (cross-PCF analyses, Fig. 3). Receptor-ligand pairs were taken from CellPhoneDB[34]. Transcriptomes for IMC populations were identified by matching IMC and scRNA-seq cells using MaxFuse. Only significant ($p < 0.05$ with false discovery rate in the

Squidpy permutation test) interactions are shown. Values are interaction scores as calculated by Squidpy. In significant sender interactions (blue), the lymphocyte expresses the ligand, and the interacting cell expresses the receptor. In significant receiver interactions (green), the lymphocyte expresses the receptor, and the interacting cell expresses the ligand. Source data are provided as a Source Data file.

cell populations (Fig. 4B), suggesting that CD4[+] TEMRA and CD8[+] TEMRA cells may share similar topological interaction profiles within VS tumours. Together, these interactions suggest the active recruitment of CD8[+] and CD4[+] T cells into the TME of VS through chemotactic and extravasation pathways, particularly in Antoni A regions, and extensive regulatory networks to limit antitumoral responses across both histomorphic niches.

**Derivation of cellular neighbourhoods reveals distinctive spatial organisation and immune environments across histomorphic niches in NF2 SWN-related vestibular schwannoma**
To further interrogate the topography of the NF2 SWN VS Antoni A and B TMEs, we deconstructed the tumour ROIs into cellular neighbourhoods (CNs). CNs are collectively defined by both the identity and proportionality of clustering cells and tumour stroma within a tissue space, and the derivation of CNs allows for visualisation of distinct subniches within heterogeneous tissues, such as tumours. To this end, we quantified CNs using CellCharter[36]. We initially computed CNs across all ROIs, assimilating both Antoni A and B regions. Whilst this identified a statistically optimal 14 distinct CNs (Supplementary Fig. 7A and B), due to the increased number of Antoni A compared to Antoni B ROIs within our dataset, these CNs were dominantly distributed within Antoni A regions; the discrete cellular networks identified in Antoni B

ROIs (Figs. 2 and 3) were lost within this combined CN analysis (Supplementary Fig. 7C). Hence, we generated unique CN profiles for both Antoni A and B regions independent of one another. In this instance, the optimal stability was achieved at $n = 10$ clusters for both Antoni A and B regions (Supplementary Fig. 7D and E).

In Antoni A regions (Fig. 5A), we identified CNs relating to inflammatory schwannoma niches characterised by pERK and PD-L1 expression (CNs A1, A5, and A7), injury response-like schwannoma niches with co-localisation of alternatively activated-like TAMs and PanCK[+]SOX-10[+] Schwann cells (CNs A0 and A2), myeloid-rich niches (CNs A4), and T-cell rich niches illustrative of perivascular infiltration and co-localisation with TAMs (CNs A3, A6 and A8). CN A9 was primarily comprised of 'Other' cell populations, and some Schwann cells. Interestingly, some of these Antoni A CNs were mirrored in the independently derived Antoni B CNs, including inflammatory schwannoma niches (CN B8), injury response-like schwannoma niches (CNs B4 and B7), myeloid-rich niches (CN B6), and immune-rich perivascular niches (CN B2 and B9). We also identified niches unique to Antoni B regions, namely demonstrating T-cell regulation through PD-L1 and alternatively activated-like TAMs, as well as TAM proliferation (CNs B0 and B3). CNs B1 and B5 were largely comprised of the 'Other' cell populations. We also quantified the abundance of CNs across cases (Fig. 5B), which reflected the single-cell composition and heterogeneity seen in Fig. 1F.

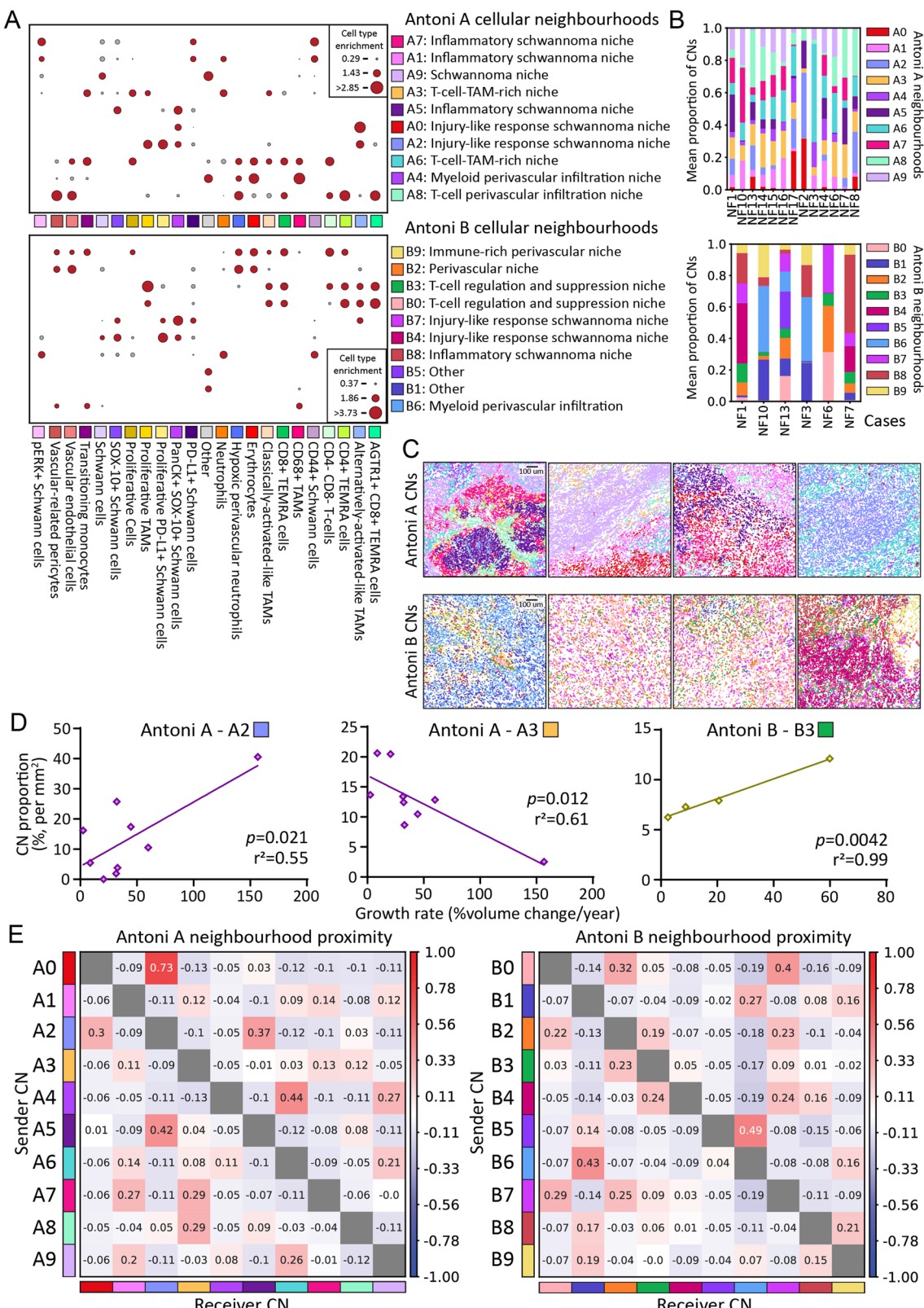

Next, we investigated the spatial layout of the different CNs in Antoni A and B regions. For the most part, Antoni A regions were comprised of similar CN densities relating to schwannoma-dominant or immune cell-dominant identities, and appeared to occupy distinct tissue regions as shown in the representative images in Fig. 5C. This would suggest that Antoni A regions had a high level of organisation. In contrast, Antoni B regions had more varied CN profiles, where some

CNs (particularly CNs B4, B6 and B8) dominated the cellular landscape of these regions. Spatially, Antoni B CNs were more inter-dispersed across the tissue regions (Fig. 5C) compared to Antoni A CNs, indicating there is less organisation of CNs in Antoni B regions compared to Antoni A regions. This is particularly true for CNs B0 and B3, suggesting that T-cell regulation in these regions occurs through discrete interactions throughout the tissue as opposed to specific focal

**Fig. 5 | Cellular neighborhood analysis identifies discrete microenvironmental regions across histomorphic niches in NF2 SWN-related vestibular schwannoma. A** Dot plot of the individually derived spatial clusters (referred to as cellular neighbourhoods, CN) for Antoni A regions and Antoni B regions. Optimal cluster stability for both groups was 10. Cells were clustered using Gaussian mixture modelling. Node size indicates relative cell type enrichment of each cell across generated CNs (red = high enrichment, grey = low enrichment) (n = 13). **B** Relative abundance of both Antoni A and Antoni B CNs across cases (n = 13).
**C** Representative spatial mapping of CNs across Antoni A and Antoni B regions.

Each image is representative on an independent case to highlight intertumoral heterogeneity (CN colour key as in **A**, **B**). **D** Significant correlations between CNs and tumour growth rate (n = 13). Correlations were assessed by either Pearson correlation coefficient or Spearman correlation. Line of best-fit was generated by simple linear regression analysis. **E** CN proximity heatmaps depicting the spatial proximity of each CN to every other CN. Y-axis indicates sender communication; x-axis indicates receiver communication. Diverging scale bar denotes proximity enrichment, with red specifying high proximity, and blue specifying low proximity. Source data are provided as a Source Data file.

aggregates. Given the distinct identities and apparent organisation of CNs in *NF2* SWN VS, we questioned if the relative abundance of CNs correlated with patient clinical parameters, namely tumour growth rate. We found that the relative abundance of CNs A2 (injury-like response) and B3 (T-cell suppression) positively correlated with tumour growth, whilst the relative abundance of CN A3 (T-cell and TAM enrichment) negatively correlated with tumour growth (Fig. 5D). No other CNs significantly correlated (positively or negatively) with tumour growth (Supplementary Fig. 7F).

Finally, we assessed the neighbourhood proximity between CNs (Fig. 5E) within the two histomorphic niches, as communication between neighbourhoods likely influences biological functions within the VS TME. In both Antoni A and B regions, it appeared that CNs of similar compositon were spatially close with each other. CN A0 had the strongest proximity to CN A2 in Antoni A regions, both of which exhibited injury response-like processes. The inflammatory tumour niches (CNs A1, A5 and A7) were closely associated with immune-rich niches (namely CNs A3, A6 and A8), as well as other inflammatory tumour niches, suggesting an inflammation-driven recruitment of immune cells within these regions. In Antoni B regions, we saw similar preferential proximity between the injury response-like niches (CNs B4 and B7), with these niches also proximally associated with niches involving T-cell suppression and TAM proliferation (CN B0 and B3). The latter CNs were also enriched for alternatively activated-like TAMs, whereby these TAMs likely support responses involving wound repair (Fig. 5E). T cells in Antoni B regions appeared to cluster within these suppressive niches (Fig. 5A), however, some resided within CN B9 and proximally co-localised with CN B6 perivascular niches (Fig. 5E). Collectively, our analyses suggest that Antoni A and B regions demonstrate a degree of spatial organisation with CNs associated with inflammation and injury-like responses conserved across both regions, however, Antoni B regions appear less organised than Antoni A regions, which may drive formation of T-cell suppressive niches across these tissue regions.

## Bevacizumab treatment significantly increases the expression of CD44 on Schwann cells, and reduces AGTR1 expression on T cells, in *NF2* SWN-related vestibular schwannoma

VS tumours in individuals with *NF2* SWN are heterogeneous both clinically and therapeutically, and as we have demonstrated this heterogeneity extends to the histopathological, cellular and spatial levels, highlighting potential for improved patient stratification and better management of the disease. To this end, we investigated the effects of bevacizumab treatment on the TME of VS from 3 patients where bevacizumab failed to control tumour growth, to enable identification of potential mechanisms leading to bevacizumab failure, and alternative modes of therapeutic intervention. These 3 cases were from *NF2* SWN patients where bevacizumab treatment was stopped prior to resection (outlined in Table 1). For this analysis, we focused only on Antoni A ROIs, as Antoni B ROIs could not be represented in the bevacizumab-treated group due to apparent low abundance of Antoni B regions within the three neuropathologist-annotated cases. Of note, tumours from patients with efficacious bevacizumab treatment could not be included in this part of the study, as tumours are not resected when treatment is effective.

Representative images of canonical Schwann cell, macrophage, T-cell, and stromal cell markers and the spatial mapping of identified cell populations between treatment naïve and bevacizumab-treated cases are shown in Fig. 6A. Interestingly, CD44+ Schwann cells were significantly more abundant, but AGTR1+ CD8+ TEMRA cells were significantly less abundant, in bevacizumab-treated cases compared to treatment-naïve cases (Fig. 6B). No significant differences were seen between the other cells types between treatment-naïve and bevacizumab-treated groups (Supplementary Fig. 8A-D). We also compared the ratio of classically activated-like TAMs to alternatively activated-like TAMs between the two groups, and found that alternatively activated-like TAMs were significantly less abundant in both treatment-naïve and bevacizumab-treated cases (Fig. 6C). We next compared the CN composition between treatment naïve and bevacizumab-treated cases, mapped in Fig. 6D; we did this by re-generating the Antoni A CNs with the additional bevacizumab cases included. The CNs from the combined naïve and bevacizumab analysis (Fig. S8E) were very similar to the naïve-only CNs (Fig. 5A), characterised by microenvironmental processes relating to inflammation, injury-like responses, and immune enrichment and infiltration (Fig. 6D, Supplementary Fig. 8E). When assessing the proportionality of CNs between naïve and bevacizumab cases (Fig. 6E), CNs Bev-0, 1 and 9 were noticeable different, with CN Bev-9 being significantly more abundant in bevacizumab than naïve cases (Fig. 6F). This CN was enriched for CD44+ Schwann cells, which parallels the significant increase in CD44+ Schwann cells in bevacizumab-treated cases (Fig. 6B). Treatment with bevacizumab led to a trend in lower proportions of CNs associated with vasculature and the perivascular niche (CNs 2, 3 and 7) but this was non-significant (Fig. 6D). Moreover, there were no qualitative histological differences in the perivascular niche in bevacizumab-treated and treatment-naïve *NF2* SWN VS cases when assessing the pericyte marker PDGFRβ (Supplementary Fig. 8F). Taken together, these analyses suggest that bevacizumab-treated tumours from patients who experienced bevacizumab failure are enriched for Schwann cell populations that may indicate ECM-related remodeling processes, as well as potential changes to the immune landscape of VS.

## Discussion

In this study, we have employed high-dimensional imaging to analyse in detail the spatial architecture of *NF2* SWN-related VS, highlighting that these tumours exhibit a complex topology with tumour-rich and immune cell-rich zones across histomorphic niche Antoni A and B regions. Importantly, beyond diagnostic value, the differing biology of Antoni A and Antoni B regions of *NF2* SWN VS tumours has not been previously explored in detail until now.

By IMC analyses, we identified 23 distinct cell populations relating to different phenotypes and activation states of Schwann cells, myeloid cells, and lymphocytes, as well as tumour stroma, across the histopathological landscape of VS. This further reinforces the complexity and cellular heterogeneity of these tumours. Nevertheless, whilst the bespoke IMC antibody panel used in this study enabled robust identification of the breadth of cell populations within *NF2* SWN VS tumours, which was supported by orthogonal MaxFuse scRNA-seq integration from a separate sample set, a limitation of the study is that the 40 antibody panel did not provide significant depth of activation

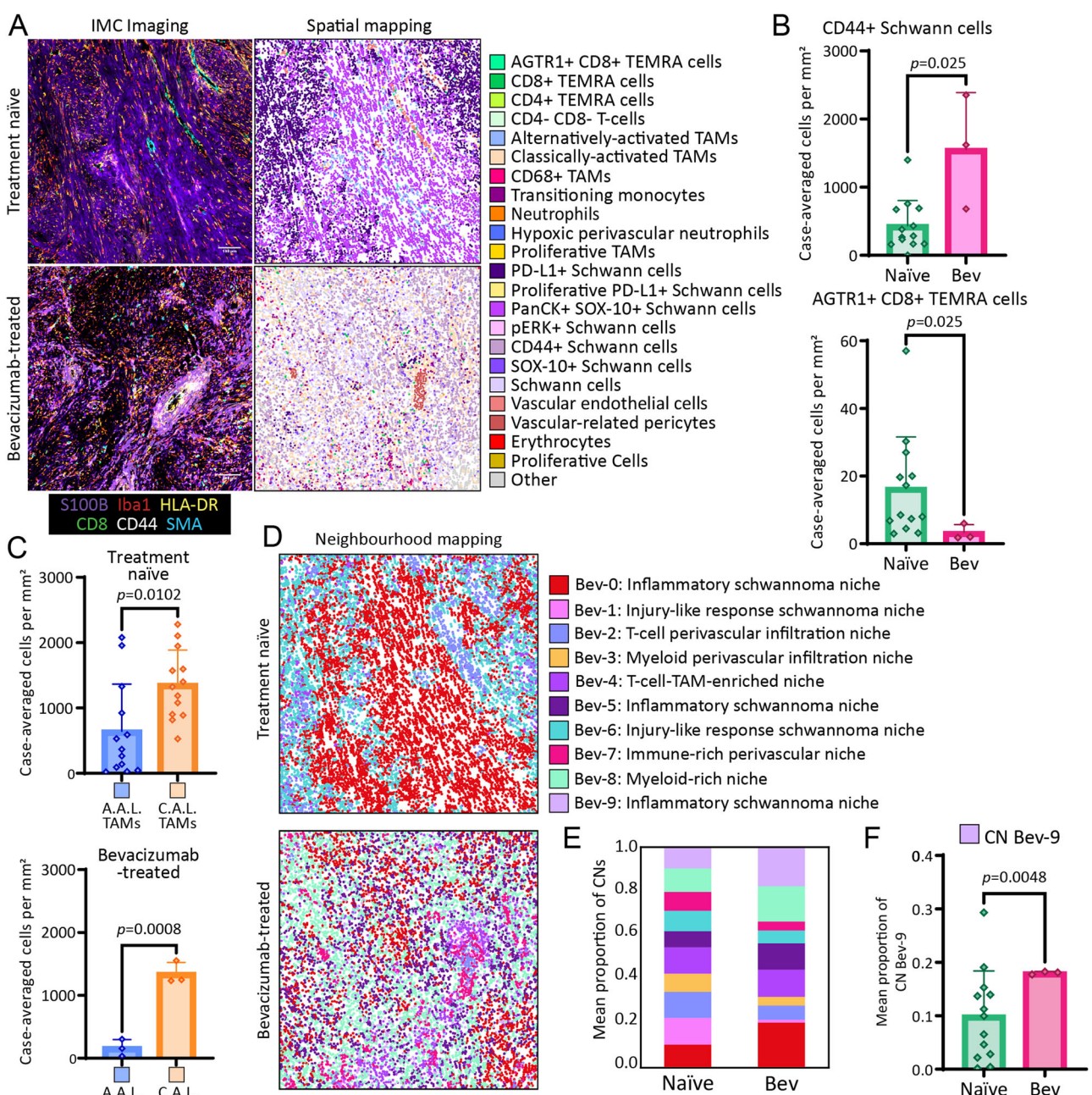

**Fig. 6 | IMC analyses highlight the influence of bevacizumab treatment on the microenvironment in NF2 SWN-related vestibular schwannoma. A** (left) IMC visualisation of core tumour and immune-related markers across treatment naïve and bevacizumab-treated groups. Markers used: S100B (purple), Iba1 (red), CD8α (green), CD44 (white), HLA-DR (yellow) and SMA (cyan). (right) Spatial seaborn maps of all populations across treatment naïve and bevacizumab-treated groups. Scale bar indicates 100 μm. **B** Abundance of significantly different cell populations across treatment naïve (*n* = 13) and bevacizumab-treated (*n* = 3) groups, tested by Mann Whitney U test. The remaining non-significant population abundances by treatment status can be found in Fig. S8A-D. **C** Comparison of alternatively activated-like TAMs versus classically activated-like TAMs across treatment naïve (*n* = 13) and bevacizumab-treated (*n* = 3) groups. Treatment-naïve tested by two-tailed Mann Whitney U test, and bevacizumab-treated tested by two-tailed unpaired *t*-test. **D** Visualisation of bevacizumab-inclusive CNs across treatment naïve and bevacizumab-treated groups. Dot plot of spatial clusters can be found in Fig. S8E. **E** Mean proportion of CNs across treatment-naïve (n = 13) and bevacizumab-treated (*n* = 3) groups. **F** Comparison of CN Bev-9 between treatment-naïve and bevacizumab treated groups, tested by two-tailed Welch's t-test. Bev = bevacizumab. (**B**, **C**, **F** result are the mean of the group + SD). Source data are provided as a Source Data file.

state and function of each cell type, and precise identification of diverse cellular subsets was challenging. Next-generation spatial transcriptomics technologies are, therefore, required to obtain more detailed spatial mapping and cell contexture within *NF2* SWN VS tumours. We also acknowledge that our IMC analyses was limited to a relatively small sample size – due to the rarity of the *NF2* SWN syndrome and the standard of care treatment for *NF2* SWN VS, where

surgery is the last resort. This may impact the generalisability of our results to VS tumours with different clinical features (related to growth rate and symptoms).

It is well established that TAMs, along with Schwann cells, are the most abundant cell type within both sporadic and *NF2* SWN-related VS[25,37]. Our study indicates that classically activated-like -pro-inflammatory TAMs are significantly more abundant than alternatively

activated-like TAMs in *NF2* SWN-related VS. However, a limitation of our study is that our IMC panel did not allow definitive identification of canonical polarised macrophage states. Indeed our analyses, in agreement with other studies, support the view that macrophages exist on a spectrum of activation states within VS tumours[30]. Whilst alternatively activated-like macrophages are perceived as tumourogenic by promoting immunosuppression and angiogenesis, classically activated macrophages may also drive immune cell recruitment, potentially contributing to tumour growth, as macrophages constitute a high proportion of VS tumour mass[24,37], and promote the symptoms associated with VS[38]. The inflammatory mediators secreted from the TME likely drives recruitment of monocytes into the tumour and support the TAM-rich identity of VS. Indeed, previous immune profiling studies in VS have shown that schwannoma cells and TAMs secrete inflammatory factors, such as interleukin (IL)-2, IL-6 and CCL20, and that increased levels of circulating monocytic chemokines are observed in plasma samples from growing VS cases[39,40]. Supporting this, we have shown that classically activated-like TAMs and transitioning monocytes have extensive interaction networks in Antoni A regions, namely with other myeloid cells, vasculature and some Schwann cell populations. Furthermore, our neighbourhood analyses demonstrated the existence of prominent perivascular networks that co-localise with inflammatory tumour regions. As such, our data suggests that there is active recruitment of monocytes through the perivascular niche in VS, which transition into classically activated-like TAMs and promote tumour inflammation in Antoni A regions. This poses the question as to whether VS growth is driven by inflammation or clonal expansion of neoplastic Schwann cells, or potentially both?

The direct cell interactions between classically activated-like TAMs and transitioning monocytes with vascular-related cells appeared lost in Antoni B regions within our analyses. The apparent disruption to direct cell–cell interactivity within the perivascular niche of Antoni B regions may be a consequence of degenerative or pathological changes to tumour vasculature, such as hyalinisation[41]. Antoni B regions have previously been proposed to result from degeneration of Antoni A regions, given their pathological similarities to Wallerian retrograde degeneration seen in traumatic nerve injury[42]. Together with our data, this suggests that the incidence of these distinct histomorphic niches may relate to spatial changes in inflammatory responses, and potentially ECM deposition, within the TME of VS.

The concept that Schwannomas arise as a consequence of injurious stimuli to peripheral nerves is evidenced by recent single-cell transcriptomic and epigenetic studies where nerve injury-like states were discovered, and characterised by myeloid cell infiltration[30,43]. Our analyses further support this finding, whereby we illustrate that alternatively activated-like TAMs significantly interact with PanCK+ SOX-10+ Schwann cells in both Antoni A and B regions; cell populations expressing both PanCK and vimentin (which is present on all Schwann cell populations we identified) are associated with wound healing responses[44], which mirror the function of alternatively activated-like TAMs. Our neighbourhood analyses also highlight these injury-like niches along with PD-L1+ Schwann cells across both Antoni A and B regions. Taken together, this suggests that injury-like niches in VS are conserved across histomorphic regions, and likely support the polarisation and maintenance of alternatively activated-like TAMs in a classically activated-like TAM-dominated landscape.

Given the importance of T cells in other tumour types as key immunotherapeutic targets[45,46], we also interrogated the positioning and proximity of both CD4+ and CD8+ T cells within *NF2* SWN-related VS. We identified 4 distinct T cells subsets, most of which displayed TEMRA phenotypes, which are associated with end-stage terminal differentiation[47]. Double negative (CD4-CD8-) T cells identified in this study have also been observed in other diseases and cancers[48], suggesting that the population is unlikely to be an imaging artefact. Whilst

these cells are regarded as being potently cytotoxic, they are also defined as having poor proliferative capacity, increased apoptosis and senescence signaling[49]. One subset of TEMRA cells expressed angiotensin II type 1 receptor (AGTR1), which has been implicated in the regulation of T-cell expansion, differentiation and function during *Plasmodium* infection[50]; the role of AGTR1 on CD8+ T cells in VS remains to be tested. The identified T cells exhibited unique spatial interactions and networks between Antoni A and B regions in VS. T cells within Antoni A regions strongly interacted with other T cells and myeloid cells (in particular classically activated-like TAMs and transitioning monocytes), as well as vasculature, but did not appear to interact with Schwann cells. This indicates that T cells may be sequestered in the TAM-enriched areas in Antoni A regions and are unable to directly target tumour cells. It has been noted that TAMs can form long-lasting interactions with T cells and impede their ability to interact with tumour cells, which directly limits the efficacy of anti-PD-1 therapies[51].

From our receptor-ligand analyses, it is possible that secretion of chemokines (including CCR5 ligands) by classically activated-like TAMs may drive the significant co-localisation between T cells (expressing CCR5) and TAMs in VS, establishing a T-cell exclusionary niche in Antoni A regions[52]. Interestingly, we observed that T-cell positioning was altered within Antoni B regions, where some T cells predominantly interacted with PD-L1+ Schwann cells, and receptor-ligand analyses predicted distinct T cell and schwann cell interactions through L1CAM, which has been shown to promote a suppressive and exhaustive phenotype in effector T cells in pancreatic cancer[53]. Furthermore, our neighbourhood analyses illustrate that CD8+ and CD4+ T cells exist within niches with PD-L1+ Schwann cells and alternatively activated-like TAMs in Antoni B regions, both of which are likely detrimental to T-cell functionality. Our receptor-ligand analyses also predicted that a variety of other suppressive pathways operate within the T-cell compartment across both Antoni A and B regions, including osteopontin (SPP1), TGFβ, SIGLEC-10 and KLRB1[54–58]. Although the PD-1/PD-L1 pathway was not identified in our predicted receptor-ligand interactions, this is potentially due to the sequencing depth of the scRNA-seq studies used for MaxFuse and CellPhone DB analyses[59], and the fact that *PDCDL1* (encoding PD-L1) appears lowly expressed in many scRNA-seq datasets[60–65]. A better understanding of how these spatially-dependent responses are orchestrated and maintained in VS could be leveraged to enable maximal T-cell-mediated tumour killing, through the potential disruption of TAM sequestration in Antoni A regions, and the blockade of inhibitory signaling provided by PD-1-PD-L1 interaction in Antoni B regions. As such, our spatial analyses highlight a clinical rationale for selected combinatorial treatment for VS tumours in *NF2* SWN.

Finally, we investigated the effects of bevacizumab treatment on the TME of *NF2* SWN-related VS. Whilst bevacizumab is not approved for *NF2* SWN by drug regulation authorities such as the FDA or MHRA, it is utilised off-label and has shown success in some *NF2* SWN patients[16,17], and is approved for treatment by NHS England in the highly specialised commissioned *NF2* service. Hence, it is important to understand why bevacizumab treatment might fail, and if alternative pathways are upregulated in bevacizumab-treated tumours that could be targeted therapeutically. Our results suggest that bevacizumab treatment may selectively change the ratio of classically activated-like TAMs to alternatively activated-like TAMs, preferentially causing loss of alternatively activated-like TAMs. Additionally, we found that bevacizumab-treated cases had significantly more CD44+ Schwann cells, yet significantly less AGTR1+ CD8+ TEMRA cells, than treatment-naïve cases. CD44, also known as the hyaluronan receptor, has various implications for cell survival, proliferation, mobility, and is often dysregulated in cancer, leading to metastasis, fibrosis and therapy resistance[66,67]. The upregulation of CD44 on Schwann cells in bevacizumab-treated VS cases may suggest the development of

fibrotic responses, which may subsequently concur with bevacizumab resistance and failure. In support of this, it has also been shown in glioblastoma that bevacizumab failure is associated with CD44 expression[68]. Interestingly, the reduction of AGTR1 expression on CD8+ TEMRA cells suggests that bevacizumab may indirectly inhibit angiotensin signaling, which is a pathway targeted by the drug losartan, which is being employed in clinical trials for VS[69,70]. Together with previous data within preclinical models of NF2 SWN VS, may indicate losartan as an alternative for NF2 SWN patients experiencing bevacizumab failure, which may be accompanied by increased matrix remodeling. However, a greater understanding of how ECM remodelling occurs in VS, and how bevacizumab treatment changes the TME in responsive tumours, is required to prove this. Notably, the bevacizumab-treated cases used in this study were largely devoid of Antoni B regions. This may further indicate that bevacizumab treatment alters the histopathological status of VS tumours; however, due to the small number of bevacizumab cases analysed, additional work will be required to confirm whether treatment directly leads to a significant loss of Antoni B regions.

In conclusion, we present a high dimensional deconstruction of NF2 SWN-related VS, where we illustrate the immune landscape is associated with clear regional differences in T-cell engagement with TAMs and Schwann cells, as well as evidence of diverse pro-inflammatory and anti-inflammatory immune and Schwann cell networks and neighbourhoods. Furthermore, we suggest potential changes in the VS TME that occur following bevacizumab treatment, and the potential link to bevacizumab failure. Although not studied here, key considerations will be how these vary in individuals with different symptoms, how they evolve in primary versus recurrent tumours, and how they differ in tumours with different pre-surgical growth rates. Nevertheless, our results give insights into the tumour features and cellular pathways that may be amenable for targeting to improve treatment of NF2 SWN-related VS.

## Methods

### Ethics
Research performed in this study complies with all relevant ethical regulations: Approved by the Health Research Authority (HRA) and Health and Care Research Wales (HCRW) for research (REC: 20/NW/0015, IRAS ID: 274046) to permit access to archived pathology specimens for research purposes in line with Medical Research Council, UK, guidelines.

### Tissue samples
16 retrospective NF2 SWN VS cases (8 male and 8 female: see Table 1 for clinical information) and 4 retrospective sporadic VS cases (used only for immunohistochemistry), in formalin-fixed paraffin-embedded (FFPE) blocks, were accessed through pathology at Salford Royal Hospital (SRH). This sample set was collected over a 4-year period (2014-2017), due to the rarity of NF2 SWN, with archived treatment naïve samples employed due to the current standard of care off-label utilisation of bevacizumab. Haematoxylin and eosin (H&E) stained slides were generated by pathology at SRH for each case and regions of interest (ROI) were selected by a neuropathologist based on identification of typical VS pathological features (Antoni A regions, Antoni B regions, immune infiltration, abnormal vasculature). Very few Antoni B regions were identified within the bevacizumab-treated cases, and as such no specific Antoni B regions of interest could be annotated for inclusion into the TMA from these cases. Once identified, 2mm² cores of each ROI were removed from each block and reformatted into tissue microarrays (TMA) through collaboration with The Christie Hospital and Cancer Research UK (CRUK) Manchester Institute. Each incorporated core in the TMA was revalidated by a second independent neuropathologist before IMC staining and data acquisition.

### H&E staining and imaging
Case blocks/TMAs were sectioned at 5μm thickness using a microtome. The 5μm tissue sections were deparaffinised in xylene for 5 min and rehydrated through an alcohol gradient (100–70% ethanol, 1 min each, then into water). Hydrated sections were incubated in haematoxylin for 3 min, followed by washing under running water for 2 min. Sections were then dipped in 1% hydrochloric acid-ethanol solution for 10 s, followed by washing under running water for 3 min. Sections were then blued using Scott's tap water for 20 s, followed by washing under running water for 5 min. Next, sections were submerged in 70% ethanol for 10 s, before being counterstained with Eosin Y for 30 s. Sections were dehydrated through an alcohol gradient (70–100% ethanol, 30 s each), then xylene for 3 min. Slides were covered slipped with DPX mountant. H&E-stained slides were imaged on the Olympus VS200 Slide Scanner (Olympus LifeScience) and digitally visualised using CaseViewer (v2.4, 3DHISTECH).

### IMC antibody conjugation, optimisation and validation
The IMC antibody panel used for this study is outlined in Table 2: sources of all antibodies are outlined in Supplementary Table 1. Antibodies were initially tested by immunofluorescence (IF) on several positive control tissues (such as secondary lymphoid tissues and tumour samples). Antibodies were conjugated to metal isotopes using the MaxPar Multimetal Labelling Kit (Standard BioTools), whereby metal isotopes were loaded onto chelating polymers, and subsequently bound to the antibody. Antibodies conjugated to platinum isotopes underwent a different conjugation protocol[71], whereby they did not require chelating polymer due to the intercalative nature of cisplatin. Final concentrations of conjugated antibodies were determined via Nanodrop (Thermofisher), followed by reconstitution with Antibody Stabilisation Solution (Candor Bioscience, Germany), and stored at 4 °C.

### IMC staining
Staining for IMC was performed according to Standard BioTools optimised protocols for FFPE tissues[72]. TMA blocks were sectioned at 5μm thickness with a microtome, deparaffinised in fresh xylene for 10 min, and rehydrated through an ethanol gradient (100–50%, 1 min per grade), and placed in distilled water. Next, tissue sections underwent antigen retrieval by incubation at 96 °C in Tris-EDTA (pH 8.5) for 30 min. Once cooled to 70 °C, tissue sections were washed in PBS (0.05% Tween) and blocked with 3% bovine serum albumin (BSA) for 45 min at room temperature (RT). Next, sections were incubated with diagnostic-grade anti-PD-L1 for 1 h at RT, washed, and a secondary anti-rabbit IgG-167Er antibody was added for 1 h at RT. Slides were then washed and then incubated with the remaining IMC antibodies (dilutions outlined in Table 2, diluted in PBS and 0.5% BSA) and incubated overnight at 4 °C. The following day, slides were washed with 0.1% Triton X-100 PBS, and incubated with Cell-ID Intercalator-Iridium (1:400 in PBS, Standard BioTools), for 30 min at RT, washed with deionised water (Merck Millipore), and dried overnight.

### IMC image acquisition
Raw IMC images were acquired using a Hyperion imaging cytometer coupled with a Helios time-of-flight mass cytometer (CyTOF, Standard BioTools). The previously neuropathologist-defined ROIs (Antoni A and B regions) were selected at 1000 × 1000 μm, and the selected tissue regions were laser ablated over several consecutive days, in a rastered pattern at 1 μm² pixel resolution, and at a frequency of 200 Hz. The resultant plume following tissue ablation was passed through a plasma source, ionising it into constituent atoms. CyTOF then differentiated the signal from each of the metal-conjugated antibodies, and images of each antibody were reconstructed based on abundance of each metal at each pixel. Images for each antibody were then exported from the raw data as TIFFs using MCD Viewer (Standard

BioTools). Acquired ROIs were screened for abnormalities during image generation and were excluded accordingly.

## Denoise clean-up and single-cell segmentation of IMC images

Raw IMC images were subjected to a denoise pipeline according to authors instructions[73]. Following denoising, single-cell data was mined from IMC images using an established protocol[74]. Stacks of TIFF images were exported from MCD files of each acquired ROI, with each channel corresponding to a metal isotope-conjugated antibody. Ilastik[75] was then applied to produce pixel probability maps that distinguished between nuclear, cytoplasmic and background pixels. These maps were then converted into cell segmentation masks, where individual cell boundaries were defined, utilising a one pixel expansion from nuclei approach for cell identification to optimise signal to noise ratio and to limit bleed through marker expression from closely localised cells, and masks were applied to each antibody channel, producing single-cell expression data for each channel, including the spatial location of each cell within the acquired ROIs. A Jaccard Index[76] (which compares the manual vs automatic segmentation of the same cells) was generated (50 random cells per region of interest) using skimage (v0.23.2) to verify the success rate of the single-cell segmentation.

## Immunohistochemistry

Slides underwent dewaxing and antigen retrieval as described under IMC staining. Endogenous peroxidases were blocked using BLOXALL Blocking Solution for 30 minutes (Vectastain, Cat. No. SP-6000), followed by blocking of endogenous Avidin and Biotin epitopes using ReadyProbes™ Avidin/Biotin Blocking Kit for 30 minutes (15 min each, Thermofisher, Cat. No. R37627). Blocking was performed with 3% BSA for 30 minutes before incubation with the primary antibody overnight at 4 ˚C. The slides were then washed three times in PBS before incubation with a biotinylated secondary antibody for 30 min. Slides were then treated with Vectastain ABC (Vectastain, Cat. No. AK-5000) for 30 min followed by incubation with Vector Blue Substrate (Vectastain, Cat. No. SK-5300) for 20 minutes. After substrate development, the above steps were repeated with Vector Red Substrate (Vectastain, Cat. No. SK-5100), when a second stain was required. For PDGFRβ, colour was instead developed using 3,3'-diaminobenzidine. PDGFRβ staining was counterstained with haematoxylin, dehydrated, and coverslipped using DPX mountant. All other stains were counterstained with DAPI, and were coverslipped using Prolong Gold (ThermoFisher). Primary antibodies used were anti-PDGFRβ (Abcam, ab69506, 42G12), anti-PD-1 (Abcam, ab137132, EPR4877(2)), anti-CD8α (BioLegend, 372902, c8/144B), anti-CD66b (BD Pharmingen, 555723, G10F5), and anti-CD11b (Abcam, ab187537, EP1345Y). Biotinylated secondary antibodies used were either horse anti-mouse (Vector Laboratories, Lot. ZF0521) or goat anti-rabbit (Vector Laboratories, Lot. ZF0809). Slides were imaged with a 3D Histech Pannoramic P25020X using a 20× objective in brightfield and fluorescence (for DAPI) and analysed using QuPath and ImageJ.

## Single-cell analyses of IMC data

The single-cell expression data was interrogated in Python using packages devised to analyse single-cell data (Scanpy v1.9.3[77]) and spatial data (Squidpy v1.2.3[78] and ATHENA v0.1.0[79]). Mean cell intensity of each marker was normalised to the 99.9th percentile of its expression, and batch corrected by case using batch balanced k nearest neighbours (BBKNN) alignment[80]. Leiden clustering[81] was used to identify cell populations. Initial Leiden populations were divided into 4 main groups (Schwann cells, myeloid cells, lymphoid cells and vasculature) based on canonical markers of each group. These broad classified cell clusters were reclustered at higher resolution to identify more granular cell sub-populations (Fig. S1C), an approach performed in other high dimensional imaging studies[82]. These populations were then re-defined on the original Leiden and annotated based upon marker expression of known cell types and states. Single-cell metrics for each population were extracted for each ROI, and used to calculate cell abundance.

## Maxfuse IMC and scRNA-seq integration

We accessed a publicly available (Gene Expression Omnibus GSE216783) single-cell RNA sequencing (scRNA-seq) dataset from 11 VS patients with confirmed *NF2* mutations[30]. Frozen samples were excluded from analysis, and raw data was pre-processed using scanpy[77], including filtering poor quality cells and genes, normalising the reads per-cell, and log transforming the data. Cells from our IMC data were then matched to equivalent cells in the scRNA-seq dataset using MaxFuse[29]. Prior to matching, populations without a clear phenotype were removed from the IMC data ('Other'), and rare populations (<1.5%) were removed from the scRNA-seq dataset if they could not feasibly be matched in the IMC dataset because specific markers were not in the IMC panel (e.g. NK and mast cells). This left 659,880 IMC cells (90.24% of original total) and 104 108 scRNA-seq cells (94.8% of original dataset) to be matched using MaxFuse, allowing matching between the 4 most abundant parental populations (myeloid, schwannoma, vascular and lymphoid) to be performed. A total of 548,455 IMC cells (83.1% of cells where matching was performed) were matched with sufficient accuracy (0.3 MaxFuse score, as recommended by authors for matching spatial protein with scRNA-seq data) for further analysis.

## Spatial omics analyses

Spatial statistical analyses were performed according to the Spatial Omics Oxford (SpOOx) analysis pipeline[32]. Cross pair correlation functions (cross-PCF) were first used to measure direct cell–cell pairwise partners within 20 μm of each cell. Adjacency cell network (ACN) analyses were generated by computing pairwise statistics between combinations of cell types, 'A' and 'B'. This was done by calculating the amount of 'A' cells in contact with 'B' cells, and then subsequently calculating the proportion of 'B' cells in contact with 'A' cells within each ROI, for each cell.

## Receptor-ligand analysis

Receptor-ligand pathways through which cells potentially communicated in the *NF2* SWN VS TME were identified using receptor-ligand analysis in squidpy[78] using the CellPhoneDB v5 database[34]. We used the transcriptomes of scRNA-seq cells that were successfully matched to our IMC populations using MaxFuse (described above). We specifically assessed interactions between CD4 and CD8 cells, and the populations they were significantly spatially associated, identified through cross-PCF analyses. Receptor-ligand pairs were filtered so that at least 5% of the cells in the sender or receiver populations expressed the ligand and receptor, respectively. For each interacting pair of populations, the top 20 strongest (defined by mean expression score calculated by squidpy) and only significant ($p < 0.05$, including false discovery rate correction using Benjamini–Hochberg procedure) receptor-ligand pairs were identified, and compared across interactions populations. Only significant interactions were visualised.

## Cellular neighbourhood analyses

Derivation of cellular neighbourhoods was performed using Cell-Charter according to authors instructions[36]. Initially, cells were sorted into neighbourhood aggregates, and then clustered by Gaussian mixture modelling (GMM). To determine the optimal number of clusters, $K$, GMM clustering was ran multiple times (10 for this study) and the highest stability clustering $n$ was selected in accordance with the Fowlkes–Mallows Index (FMI). All cells were then sorted into spatial clusters. Next, proximity analysis was performed to assess the relative arrangement of spatial clusters within ROIs, indicating neighbourhood enrichment between clusters.

## Statistics

Statistical tests were performed using GraphPad Prism (v9.5.1). Data distribution was assessed by Shapiro-Wilk normality test. Differences in cell abundances between two groups (Antoni A versus Antoni B; naïve versus bevacizumab) were performed employing case averages using unpaired t-test or Mann Whitney U test. For differences between three groups (disease severity), cell abundances were compared employing case averages using a one-way ANOVA with Tukey's multiple comparisons test or Kruskal–Wallis test with Dunnett's T3 multiple comparisons test. Specific spatial statistical analyses were performed as described in each section. IMC data is from one experiment.

## Reporting summary

Further information on research design is available in the Nature Portfolio Reporting Summary linked to this article.

## Data availability

The raw and processed IMC data (images denoised using IMC-Denoise[73], segmentation masks created using an established pipeline[74], Python AnnData object in.h5ad format, and all cell- and sample-level metadata) generated in this study have been deposited in the DataDryad database [https://doi.org/10.5061/dryad.wwpzgmstv]. All other data generated in this study are provided in the Supplementary Information and Source Data are provided with this paper. The scRNA-seq data from vestibular schwannoma used in this study are available in the Gene Expression Omnibus database under accession code GSE216783. Source data are provided with this paper.

## Code availability

All computational analyses employed published resources, as outlined in materials and methods.

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

## Acknowledgements

We thank Dr. Gareth Howell and Dr. Jennifer Baron within the Flow Cytometry Core Facility at the University of Manchester for helping with the imaging mass cytometry work performed in this study. We thank Dr. Peter March in the Bioimaging Core Facility at the University of Manchester for helping with the microscopy work. We thank Professor Federico Roncaroli and Dr. Daniel du Plessis within the Department of Cellular Pathology at the Salford Royal Hospital for annotating and validating the H&E-stained slides to identify Antoni A and B regions for incorporation within the Tissue Microarrays, which were generated by Dr. Garry Ashton and team at The Christie Hospital (CRUK), Manchester. The work was funded by *NF2 BioSolutions* (PhD studentships supporting A.P.J and G.E.G), the Medical Research Council (MR/T016515/1 to D.B and K.N.C, MR/V034650/1 to K.N.C). This study has been delivered through the National Institute for Health and Care Research (NIHR) Manchester Biomedical Research Centre (BRC) (NIHR203308 to D.B and O.N.P; IS-BRC-1215-20007 to D.G.E, D.B. and O.N.P). The views expressed are those of the author(s) and not necessarily those of the NIHR or the Department of Health and Social Care. The imaging mass cytometer used within the study was purchased through a BBSRC Alert18 award (BB/S019324/1 to K.N.C.).

## Author contributions

Conceptualisation of study by A.P.J., K.N.C. and G.E.G.; Methodology by MJH and M.H.M.; Investigation performed by A.P.J., G.E.G., A.K.S. and M.H.M.; Resources provided by C.J.H., D.G.L., P.O., L.D.B., M.J.S. and A.T.K.; Software by M.J.H. and A.P.J.; Visualisation performed by A.P.J., M.J.H. and M.H.M.; Study supervised by K.N.C., O.N.P., D.B. and P.P.; Original draft of manuscript by A.P.J. and K.N.C.; Reviewing and editing of manuscript by A.P.J., K.N.C., O.N.P., D.B., P.P., D.G.E., A.T.K., M.J.S., P.O., D.G.L., C.J.H., M.J.H. and G.E.G.: Funding acquisition for study by K.N.C., O.N.P., D.B. and D.G.E.

## Competing interests

The authors declare no competing interests.

## Additional information

¹Division of Immunology, Immunity to Infection and Respiratory Medicine, Faculty of Biology, Medicine & Health, The University of Manchester, Manchester, UK. ²Geoffrey Jefferson Brain Research Centre, Manchester Academic Health Science Centre, Northern Care Alliance NHS Foundation Trust, University of Manchester, Manchester, UK. ³Division of Neuroscience, Faculty of Biology, Medicine & Health, The University of Manchester, Manchester, UK. ⁴Department of Neurosurgery, Manchester Centre for Clinical Neurosciences, Salford Royal Hospital NHS Foundation Trust, Salford, UK. ⁵Department of Pathology, The Christie Hospital, Manchester, UK. ⁶Division of Evolution, Infection and Genomics, Faculty of Biology, Medicine & Health, The University of Manchester, Manchester, UK. ⁷Department of Biosystems and Soft Matter, Institute of Fundamental Technological Research, Polish Academy of Sciences, Warsaw, Poland. ⁸These authors contributed equally: Adam P. Jones, Michael J. Haley. ⁹These authors jointly supervised this work: David Brough, Omar N. Pathmanaban, Kevin N. Couper. ✉e-mail: David.brough@manchester.ac.uk; omar.pathmanaban@manchester.ac.uk; kevin.couper@manchester.ac.uk

