## [Transparent Peer Review file · Nature Communications]

Spatial Mapping of Immune Cell Environments in NF2-related Schwannomatosis Vestibular Schwannoma

Corresponding Author: Professor Kevin Couper

Version 0:

Reviewer comments:

Reviewer #1

(Remarks to the Author)

In the study "Spatial mapping of immune cell environments in NF2-related schwannomatosis vestibular schwannoma," the immunological landscape of NF2-related vestibular schwannomas (VS) is comprehensively characterized using mass cytometry in Antoni A and B areas. The authors reveal not only the heterogeneity in the tumor microenvironment (TME) composition but also spatial differences between Antoni A and B regions.

The methodology is detailed and thorough, and the paper is written clearly and cohesively. I recommend accepting the paper with one major revision and some minor revisions.

Major:

- Table 1 and 2 are cropped and are missing the captions

Minor:

- L 39: typo: missing the word "are"

- L84: introduce the term interactome

- L 99 and L453: was annotation performed by one or two experienced neuropathologists? What did you do to prevent possible mistakes in annotation?

- Fig.1 E and Fig.5 C: Y-axis should be the same for better comparison

- L 397: typo: "is drives"

Reviewer #2

(Remarks to the Author)

Reviewer #3

(Remarks to the Author)

The study investigates the complex immunological networks within VS tumors in NF2-related Schwannomatosis (NF2 SWN) patients. The authors utilized IMC to analyze clinically annotated VS samples, revealing significant heterogeneity in neoplastic, myeloid, and T-cell populations across different histomorphologic niches (Antoni A and B regions). The study also examines the impact of bevacizumab treatment on the tumor microenvironment.

Key Findings

Heterogeneity in Cell Populations: The study identifies distinct populations of Schwann cells, myeloid cells, and T-cells within VS tumors. Antoni A regions are characterized by interactions between T-cells and tumor-associated macrophages (TAMs), while Antoni B regions show T-cell interactions with PD-L1+ Schwann cells.

Impact of Bevacizumab: Bevacizumab treatment alters the immune landscape by reducing alternatively-activated TAMs and increasing CD44 expression in Schwann cells, suggesting changes in matrix remodeling.

Spatial Interactions: The study highlights niche-dependent modes of T-cell regulation, with T-cells in Antoni A regions being sequestered in TAM-rich areas, limiting their interaction with tumor cells. In Antoni B regions, T-cells interact more with immunoregulatory PD-L1+ Schwann cells.

Cellular Neighborhoods: The authors deconstructed the tumor regions into cellular neighborhoods, identifying distinct microenvironmental niches associated with inflammation, injury response, and immune regulation.

Strengths

Comprehensive Analysis: The use of IMC provides a high-dimensional view of the tumor microenvironment, allowing for detailed spatial mapping of immune cell interactions.

Clinical Relevance: The findings have potential implications for developing targeted therapies for NF2 SWN-related VS, particularly in understanding the mechanisms of bevacizumab resistance.

Novel Insights: The study provides new insights into the spatial organization and immune environments within VS tumors, highlighting the importance of niche-specific interactions.

Well-Written and Organized: The manuscript is well-structured, making the complex data accessible and understandable.

Weaknesses

Sample Size: The study analyzes a relatively small number of cases (16), which may limit the generalizability of the findings.

Focus on Antoni A Regions: The analysis of bevacizumab-treated tumors focuses only on Antoni A regions, potentially overlooking important changes in Antoni B regions.

Limited Technology Scope: The study focuses solely on spatial analysis using IMC and does not incorporate other technologies like single-cell RNA sequencing, which could provide additional insights into cellular abundance and density.

Marker Limitation: The use of mass spectrometry limits the study to 40 protein markers, which may restrict the identification of immune cell subsets.

Lack of Comparative Analysis: The comparison of bevacizumab-treated samples to treatment-naïve samples does not include responders, which could provide more relevant insights into treatment efficacy.

Specific Comments

Explanation for Antoni B Regions in Bevacizumab Samples: The manuscript should include a possible explanation for the absence of Antoni B regions in bevacizumab-treated samples. Is this due to sampling issues based on H&E slides, or is it biologically relevant?

Additional Clinical Data: The authors should explore whether additional clinical data (e.g., tumor size, location, progression) correlate with the spatial data presented in the manuscript.

Vascular Interactions in Bevacizumab Samples: Given the decrease in AGTR1+ CD8+ TEMRA cells in bevacizumab-treated samples, it would be interesting to see if there are changes in interactions with vascular endothelial cells and vascular-related pericytes.

Supplementary Figure Reference: The manuscript mentions supplementary figure 6B, which does not exist in the reviewed materials. This should likely refer to supplementary figure 5E.

Ligand-Receptor Interaction Analysis: If the technology permits, including ligand-receptor interaction analysis for significant interactions identified with cross pair correlation analysis and/or adjacency network analysis would further define these interactions.

Discussion on TAMs and Treatment Resistance: The manuscript should include further discussion on how the increased ratio of classically-activated to alternatively-activated TAMs contributes to treatment resistance or changes in the tumor microenvironment in bevacizumab-treated samples.

Reviewer #4

(Remarks to the Author)

Reviewer #5

(Remarks to the Author)

In the manuscript entitled "Spatial mapping of immune cell environment in NF2-related Schwannomatosis vestibular Schwannoma", Jones et al. applied Imaging mass cytometry to 13 cases of vestibular schwannoma with NF2 mutations, to define the tumor microenvironment, revealing a different cell composition and cell-to-cell interaction in Antoni A vs Antoni B niches. The authors identified 23 different cell phenotypes, with several TAMs and Schwann cells phenotype described and being the major component of the TME in VS. The authors described several differences in cell-to-cell interaction in the Antoni A vs Antoni B areas. Additionally, several cellular neighborhoods have been identified and the reciprocal interaction have been assessed in Antoni A and B niches. Finally, the authors also compared TME of three (3) patients treated with Bevacizumab prior tumor resection and immune microenvironment of tumor treatment-naïve, identifying CD44+ Schwann cells has more abundant after Bevacizumab treatment, while AGTR1+ CD8+ TEMRA were reduced and potentially related to intratumoral remodeling. Vestibular Schwannoma with NF2 mutation is a rare disease and as such its cellular landscape is not well characterize, however the manuscript focus has been placed on the cell-cell interaction analysis performed via cross pair correlation functions and cellular neighborhood analysis done by applying CellCharter and GMM.

Strengths of this study include detailed characterization of a rare disease using spatial proteomics. This is also a limitation as the sample size is small (13 cases and 3 with prior VEGF treatment) without validation using an external dataset. Further, the quality of the clustering is impacted by marker distributions that are not representative of immune subsets shown in Figure 1c, where there are many CD3+ clusters that do not express either CD4 or CD8. One CD3+ cluster is labeled as a

TAM (light blue), while another is labeled as vascular endothelial cells (pink). CD163 is diffusely positive. This impacts all the downstream analysis and interpretation of spatial relationships. Together, with a low sample size and issues with staining and clustering, reduces the potential impact of this manuscript.

Additional points that need further clarification:

1) It seems that authors have used CD11b as marker for identification of neutrophils. However, CD11b is a marker expressed by several immune cells, including monocytes and macrophages, since it mediates immune cells migration and chemotaxis. Additionally, for this cluster there is no granzyme B expression. The only marker that can be used alone for neutrophil identification is CD66b, which is not included in this panel. Thus, it is hard to assign with high degree of certainty this phenotype based on the sole expression of CD11b. How do the authors justify the phenotype assignment? Have the authors tried other clustering algorithms which might provide a better clustering and improve the accuracy of cell phenotypes?

2) How did the authors define alternatively activated and classically activated TAMs? Alternatively-activated TAMs are actually expressing higher level of PanCK, S100B compared to CD163 and CD68 and Iba1 (which I think was used as the most reliable marker for macrophage identification, but it still presents a lower expression compared to markers used for Schwann cell). CD68 expression is null in classically-activated TAMs. It is worth teasing out cells in these clusters in order to obtain cleaner clusters/phenotypes. Alternatively, the authors should support their choice in phenotype assignment.

3) How do the authors explain the higher expression of Tim3 in Schwann cells compare to T-cells or other immune cells? While the authors have 40 markers in their panel, some of those seem to be completely disregarded throughout the entire manuscript. This is especially true for signaling markers, which are not discussed or described in the context of the appropriate cell lineage. This is limiting the biological and immunological contribution this paper might provide.

4) CD206 is used as marker for M2 polarized macrophage recognition, which has no expression in the present study. Since the authors preprocessed the raw images using IMC-denoise, do they altered the expression of this marker while cleaning the data? Usually, CD206 gives a good signal which helps in macrophages classification. Can the authors clarify this point?

5) One of the major information missing is the number of cells per clusters and the total number of cells they have in the single cell database. Can the authors provide the percentage of each cluster they have identified in the dataset and for each patient? What is the number or percentage of cell in "others" the unspecified cluster? How many cells were not classifiable based on the current panel?

6) Figure 1C: with such a large number of clusters, the authors should re-organize the heatmap such that the clusters order in the graph follows the order of clusters in the legend. This would make the interpretation and reading of the heatmap easier for the readers.

7) How many Antoni A and B region were examined in the present study? How many per patient?

8) Fig 1F: the study is examining 16 cases of VS, 13 treatment naïve and 3 post treatment. The first part of the paper seems to describe 13 naïve cases (line 97), however, in the figure 1F there are 16 cases listed. Later, they discussed the CN in naïve versus bevacizumab-treated cases. Can the authors clarify if the three treated patients were considered twice in the analysis (before and after resection)? It is confusing whether the first CN analysis and in general the first part of the paper is done on 13 cases excluding the three bevacizumab-treated cases or not, especially looking at Fig1 where there are 16 cases reported. The authors should clarify the case selection in the first part of the paper.

9) Fig4A: the legend and the dot plots are difficult to follow. Please re-organize the CNs based on their order in the legend, so it is easier to cross-check the information.

10) Last part of the paper is confusing and should be revised. The authors are referencing supplemental figures not provided (there is no Fig 6B in the supplemental material). Line 325 is referring to the wrong figure (it should 5F instead of 5E).

11) Supplementary fig 5E is describing something not reported in the actual figure. (there is no elbow plot).

12) The authors should mention the limitations of their study, which include at least the small sample size and the lack of important markers, such as PD1, especially in light of their finding and the emphasis they place in the presence of PDL1 Schwann cells, which have been described has one of the clusters with the greatest number of interactions. Thus, it would be important to have PD1 and mention this aspect as a limitation of the study.

13) Line 269: "biological processes outlined above and appeared to occupy distinct tissue regions as shown in the representative images in Fig 4C": which biological process are they referring to? This sounds unclear and the concept should be clarified as the figure 4C is a neighbor maps with several cell types.

Minor comments:

1) Abstract line 39-40: revise the sentence " Interestingly, T-cell populations associated with tumour-associated macrophages (TAMs) in Antoni A regions, seemingly limiting their ability to interact with tumorigenic Schwann cells". Did the authors meant ...are associated?

2) Line 53: "...the severity phenotype..." should be the severity of the phenotype?

3) Line 344: "greater proportion of the myeloid compartment"...the proportion has never been mentioned so provide this information is pivotal.

Version 1:

Reviewer comments:

Reviewer #2

(Remarks to the Author)

In this revised manuscript, the authors strengthen their study by incorporating external scRNAseq data and perform IHC on additional VS samples to validate their IMC cell annotation. Additionally, addition of scRNAseq data allowed for analysis of ligand-receptor interactions, which provided additional insight into the cell-cell interactions defined by their spatial analysis. While their sample size is limited, it is adequate given the incidence of NF2 VS and provides further insight into a rare disease. The authors insightfully point out that absence of surgical management for VS that respond to bevacizumab treatment precludes their ability to compare bevacizumab responders and non-responders. Therefore, comparison of bevacizumab non-responders to treatment naïve samples still provides insight into mechanisms of disease resistance. The authors also attempt to correlate some of their results with clinical features of VS. While they did not observe any correlation with immune cell populations and disease severity based on NF2 pathologic variants, this may be due to the clinical features of this cohort. Given that surgery for VS is only performed in cases where there is increased tumor growth or patients become symptomatic, the samples in this cohort likely have more severe clinical features compared slow growing tumors that were able to be observed. Thus, it will be more difficult to determine how changes in the tumor immune microenvironment contribute to progressive tumor growth. Additionally, the authors demonstrate that cellular networks associated with injury response and T-cell suppression correlated with tumor growth while cellular networks associated with enrichment of T cells and TAMs negatively correlated with tumor growth, which does provide some additional insight into disease biology.

Overall, the authors have addressed all the major concerns address in the initial review. I have no additional major concerns that need to be addressed prior to publication. I only have a few minor comments:

- Line 149: schwann cells should be capitalized
- Line 252: "was not within the IMC, panel" – comma is not in the correct location, should be "was not within the IMC panel,"
- bevacizumab and losartan are generic names of drugs so should not need to be capitalized
- Was CD Bev-0 significantly enriched in the bevacizumab treated samples? Based on the mean proportion graph in Figure 6E, there appears to be a greater difference than for the CN Bev-9, which was significantly different.

Reviewer #3

(Remarks to the Author)

Based on my review of the manuscript and the authors' responses, I believe the authors have been largely responsive to the reviewers' critiques and have made substantial improvements to the manuscript. However, there are a few areas where the authors could have been more responsive or where further clarification may be needed:

Sample size limitation: While the authors provided a detailed explanation for the small sample size due to the rarity of NF2-related schwannomatosis, they could have more directly addressed how this limitation impacts the generalizability of their findings. A more explicit discussion of this limitation in the manuscript would be beneficial.

Antoni B regions in bevacizumab-treated samples: The authors explained the absence of Antoni B regions in bevacizumab-treated samples, but they could have elaborated more on the potential biological significance of this observation in the manuscript itself.

Marker limitations: While the authors acknowledged the limitations of their 40-marker panel, they could have provided more discussion on how this might have impacted their ability to identify certain cell subsets, particularly in light of the reviewer's concerns about CD3+ clusters lacking CD4 or CD8 expression.

Cell segmentation and clustering: The authors provided a detailed response regarding their segmentation and clustering approach, including validation steps. However, they could have incorporated more of this explanation into the manuscript itself to address potential concerns about the quality of cell identification.

Neutrophil identification: While the authors performed additional IHC staining to confirm CD66b+ neutrophils, they could have more explicitly addressed the limitations of using CD11b alone for neutrophil identification in their IMC panel within the manuscript.

TAM classification: The authors provided reasoning for their classification of alternatively activated-like and classically activated-like TAMs, but they could have incorporated more of this explanation into the manuscript to justify their phenotype assignments.

Overall, the authors have made significant efforts to address the reviewers' comments and have substantially improved the manuscript. However, incorporating more of their detailed responses directly into the manuscript text would further strengthen their arguments and address potential reader concerns.

Reviewer #4

(Remarks to the Author)

Reviewer #5

(Remarks to the Author)

The authors addressed all questions and provided additional data and clarified all the points raised by the reviewers. I do not have any additional comments.

We thank the editor and reviewers for their constructive and positive comments on our manuscript. In revision, we have substantially modified our manuscript to respond to the editor's and reviewers' points, including adding additional new figures (Figure 4, Supplementary Figures 2, 3 and 6), modifying figures (Figure 5, Supplementary Figures 1, 7 and 8) and substantially altering the results and discussion sections of the manuscript. We also provide 2 figures for review purpose only. All changes in the revised manuscript are highlighted in **yellow**. We provide a point-by-point response to the reviewers' comments, below.

REVIEWER COMMENTS

Reviewer #1 (Remarks to the Author):

In the study "Spatial mapping of immune cell environments in NF2-related schwannomatosis vestibular schwannoma," the immunological landscape of NF2-related vestibular schwannomas (VS) is comprehensively characterized using mass cytometry in Antoni A and B areas. The authors reveal not only the heterogeneity in the tumor microenvironment (TME) composition but also spatial differences between Antoni A and B regions. The methodology is detailed and thorough, and the paper is written clearly and cohesively. I recommend accepting the paper with one major revision and some minor revisions.

1.1 Major:

- Table 1 and 2 are cropped and are missing the captions

Response: We thank the reviewer for identifying this error, which we have corrected in the revised manuscript.

1.2 Minor:

- L 39: typo: missing the word "are"

Response: This has been corrected in the revised manuscript.

1.3 - L84: introduce the term interactome

Response. We have introduced this within the revised manuscript (**line 84**).

1.4 - L 99 and L453: was annotation performed by one or two experienced neuropathologists? What did you do to prevent possible mistakes in annotation?

Response: H & E stained tissue sections were initially examined and annotated by one experienced neuropathologist. This allowed us to select tumour regions of interest (Antoni A and Antoni B histomorphologic zones) for inclusion into the tissue microarray (TMA). Following creation of the TMA, new H & E stained sections were created for the incorporated tumour cores, which were subsequently reanalysed by a second neuropathologist, to confirm the classification of the regions of interest (ROIs) for downstream analysis. We have added text in the revised manuscript (**lines 100-101, 581-582**) to provide this additional information.

1.5 - Fig.1 E and Fig.5 C: Y-axis should be the same for better comparison

Response: This has been corrected in the revised manuscript

1.6 - L 397: typo: "is drives"

Response: This has been corrected

Reviewer #3 (Remarks to the Author):

The study investigates the complex immunological networks within VS tumors in NF2-related Schwannomatosis (NF2 SWN) patients. The authors utilized IMC to analyze clinically annotated VS samples, revealing significant heterogeneity in neoplastic, myeloid, and T-cell populations across different histomorphologic niches (Antoni A and B regions). The study also examines the impact of bevacizumab treatment on the tumor microenvironment.

Key Findings

Heterogeneity in Cell Populations: The study identifies distinct populations of Schwann cells, myeloid cells, and T-cells within VS tumors. Antoni A regions are characterized by interactions between T-cells and tumor-associated macrophages (TAMs), while Antoni B regions show T-cell interactions with PD-L1+ Schwann cells.

Impact of Bevacizumab: Bevacizumab treatment alters the immune landscape by reducing alternatively-activated TAMs and increasing CD44 expression in Schwann cells, suggesting changes in matrix remodeling.

Spatial Interactions: The study highlights niche-dependent modes of T-cell regulation, with T-cells in Antoni A regions being sequestered in TAM-rich areas, limiting their interaction with tumor cells. In Antoni B regions, T-cells interact more with immunoregulatory PD-L1+ Schwann cells.

Cellular Neighborhoods: The authors deconstructed the tumor regions into cellular neighborhoods, identifying distinct microenvironmental niches associated with inflammation, injury response, and immune regulation.

Strengths

3.1 Comprehensive Analysis: The use of IMC provides a high-dimensional view of the tumor microenvironment, allowing for detailed spatial mapping of immune cell interactions.

Clinical Relevance: The findings have potential implications for developing targeted therapies for NF2 SWN-related VS, particularly in understanding the mechanisms of bevacizumab resistance.

Novel Insights: The study provides new insights into the spatial organization and immune environments within VS tumors, highlighting the importance of niche-specific interactions.

Well-Written and Organized: The manuscript is well-structured, making the complex data accessible and understandable.

Response: We thank the reviewer for their positive assessment of our manuscript.

3.2 Weaknesses

Sample Size: The study analyzes a relatively small number of cases (16), which may limit the generalizability of the findings.

Response: We acknowledge the reviewer's comment. NF2 schwannomatosis (NF2-SWN) is a rare disease (affecting 1 in 40000 people) and although Manchester is a major centre for treatment of NF2-SWN patients, there are few VS surgeries on NF2-SWN patients each year (170 over a 30 year period, equating to less than 6 per year). The standard of care treatment for

NF2-SWN patients now involves active surveillance of tumours, off label usage of Bevacizumab, with surgery only performed when eventually necessary to preserve hearing and quality of life. This has further reduced the number of surgeries in the last 10 years, limiting the number of cases (particularly treatment naïve cases) collected for research. Consequently, the sample set utilised in this study was collected over a 4 year period (2014-2017) and represents a good number of *NF2*-SWN VS tumours, given the epidemiology of the disease. As such, we were unable to perform analyses on the numbers of cases as described in other imaging mass cytometry cancer studies, such as for breast or lung cancer or glioblastoma with significantly higher prevalence.

Although we investigated a relatively small number of cases, our analyses – using established and robust methodologies applied and tested in various other disease and cancer contexts (as outlined in materials and methods of the manuscript) – identified significant and reproducible spatial features within *NF2*-SWN VS tumours. This emphasises that as long as sufficient ROIs are run per case and enough single cells are segmented and captured, robust data can be obtained using imaging mass cytometry and high dimensional imaging with a relatively small number of samples. Indeed, relatively small sample sets, equivalent in number to that employed in our study, have also been utilised in other studies, providing in depth insight into COVID, tissue architecture and cancer (Schabenland *et al* Immunity 2021; Rendeiro *et al* Nature 2021, Hickey *et al* Nature 2023, Ji *et al* Cell 2020).

Importantly, to mitigate any potential bias in our analyses due to individual cases, we also presented all our results as case averaged data (rather than ROI averaged data), where each case represented one biological sample within the group. This controlled for instances where different numbers of ROIs were analysed in different cases.

As described below in response to reviewer 3 point 4, in revision we have verified that the cell populations identified through our IMC analyses are represented in an orthogonal scRNA-seq dataset from 11 VS tumours with *NF2* mutation (**Fig S3**). We believe this adds substantial additional weight to the generalisability of our observations

We have added text into the revised manuscript to address this comment and the size of our dataset, as well as to provide consideration of the generalisability of our findings (**lines 439-441 and 570-572**).

3.3 Focus on Antoni A Regions: The analysis of bevacizumab-treated tumors focuses only on Antoni A regions, potentially overlooking important changes in Antoni B regions.

Response: We agree with the reviewer that we need to provide more clarity on this point.

There were very few Antoni B regions within the bevacizumab-treated cases, and as such no specific Antoni B regions of interest could be annotated by our neuropathologist for inclusion into the TMA from these cases. We have added text into the revised manuscript (**lines 390-393, 549-552 and 576-578**) to indicate this point and to emphasise that additional work will be required to understand if bevacizumab treatment lowers the incidence and relative abundance of Antoni B regions within *NF2*-SWN VS tumours. However, as described below (reviewer 3 point 6), it will not be possible to address if the treatment efficacy of bevacizumab correlates with Antoni B loss, given tumours are not removed from patients when bevacizumab is efficacious.

3.4 Limited Technology Scope: The study focuses solely on spatial analysis using IMC and does not incorporate other technologies like single-cell RNA sequencing, which could provide additional insights into cellular abundance and density.

Response: Although IMC is extremely powerful for identifying, mapping and understanding the contexture of cells within a tissue or tumour environment, we agree with the reviewer that confirmation of results using different technologies and approaches is useful.

In response to this comment, we have integrated our IMC data with published scRNA-seq datasets of sporadic VS tumours (11 tumours, all exhibiting NF2 mutations; Barret *et al* Nat Comms 2023: PMID: 38216553). Unfortunately, as we utilised retrospective FFPE samples in our IMC analyses, we did not have suitable samples to perform matching scRNA-seq and IMC analyses on the same cases.

Through the use of MaxFuse (Chen *et al* Nat Biotechnol 2024: PMID: 37679544), we were able to attempt matching of lymphoid, myeloid, schwannoma and vascular cells (collectively, 90.4% of the cells in our IMC dataset) with corresponding cells in the scRNA-seq dataset. In total, 85.7% of these cells were accurately matched between modalities (with recommended MaxFuse accuracy >0.3). These integrative analyses outlining the Maxfuse approach, the matching of IMC-defined and scRNA-seq-defined cells via UMAP, the expression profile of genes (corresponding to antibody protein targets in our IMC panel) across cell population regions within the integrated UMAP, and the heat map representing gene expression signatures within the MaxFuse matched IMC and scRNA-seq populations, are presented in Fig S3 and are described in the revised manuscript (**lines 153-158, 433-441**).

Collectively, we believe that this novel multi-modal analysis has orthogonally validated our IMC populations. It has also enabled more in-depth analyses, for example, receptor-ligand analyses (as described below in response to reviewer 3 point 11), which has significantly improved the impact of our study

3.5 Marker Limitation: The use of mass spectrometry limits the study to 40 protein markers, which may restrict the identification of immune cell subsets.

Response: We acknowledge this point. Although IMC (and equivalent technologies such as codex/phenocycler) are providing crucial new insight into the organisation of tumour microenvironments and tissues, they are limited by number of antibodies that can be employed, and a priori knowledge to guide antibody panel design.

We spent a substantial amount of time attempting to optimise our IMC antibody panel so that it allowed us to characterise and sub-cluster different known and relevant cell populations within NF2-SWN VS tumours. This resolved different myeloid cell, Schwann cell, T cell and vascular-associated cell populations, which we have now verified within the integrative multi-modal MaxFuse scRNA-seq analyses, as described in response to the preceding comment. Nevertheless, we add text into the revised manuscript to highlight this limitation and that utilisation of new technologies, such as MERSCOPE, COSMX or VISUM HD are required to obtain more depth of spatial single cell mapping and contexture in NF2-SWN VS tumours (**lines 433-439**).

3.6 Lack of Comparative Analysis: The comparison of bevacizumab-treated samples to treatment-naïve samples does not include responders, which could provide more relevant insights into treatment efficacy.

Response: We agree with the reviewer that this is a limitation. We would have liked to have included tumours from bevacizumab responders in our analyses; however, it is not possible to obtain samples from people currently responding to bevacizumab, as the tumours are not resected when the treatment is effective. We have added text into the revised manuscript (**lines 393-395**) to address this point.

Specific Comments

3.7 Explanation for Antoni B Regions in Bevacizumab Samples: The manuscript should include a possible explanation for the absence of Antoni B regions in bevacizumab-treated samples. Is this due to sampling issues based on H&E slides, or is it biologically relevant?

Response: We address this point above (reviewer 3 point 3).

3.8 Additional Clinical Data: The authors should explore whether additional clinical data (e.g., tumor size, location, progression) correlate with the spatial data presented in the manuscript.

Response: We thank the reviewer for this suggestion. We have performed additional analyses to investigate whether the relative abundance of the identified cellular neighbourhoods (CNs) corresponded with tumour growth rate. Interestingly, we observed a significant positive correlation between tumour growth and CN A2 in Antoni A regions, which is associated with injury-like responses. Conversely, we saw a significant negative correlation between tumour growth rate and CN A3, which is characterised by immune enrichment of T-cells and macrophages. In Antoni B regions, there was a significant positive correlation between tumour growth and CN B3, which reflected T-cell suppression. We have provided these correlative analyses in **Fig 5D**, **Fig S7F** and are described (**lines 355-361**) in the revised manuscript.

3.9 Vascular Interactions in Bevacizumab Samples: Given the decrease in AGTR1+ CD8+ TEMRA cells in bevacizumab-treated samples, it would be interesting to see if there are changes in interactions with vascular endothelial cells and vascular-related pericytes.

Response: We thank the reviewer for this good suggestion. As we did not have antibodies to pericyte markers (NG2, PDGFR β) in our IMC panel, we performed IHC analyses on existing TMA slides, which included treatment naïve and bevacizumab-treated samples. Qualitatively, we did not observe substantial changes in PDGFR β staining within Bevacizumab and treatment naïve NF2-SWN VS tumours. We provide this new data within **Fig S8F** and describe it on lines **416-419** in the revised manuscript. We also provide additional evaluation of the relative abundance of vascular and perivascular niche cellular neighbourhoods in bevacizumab and treatment naïve NF2-SWN VS cases (**lines 414-416**).

3.10 Supplementary Figure Reference: The manuscript mentions supplementary figure 6B, which does not exist in the reviewed materials. This should likely refer to supplementary figure 5E.

Response: We apologize for this error. We have corrected this in the revised manuscript

3.11 Ligand-Receptor Interaction Analysis: If the technology permits, including ligand-receptor interaction analysis for significant interactions identified with cross pair correlation analysis and/or adjacency network analysis would further define these interactions.

Response: We agree that such analyses would substantially improve our manuscript. Receptor-ligand interaction analysis is, unfortunately, not feasible directly within our IMC datasets (due to the insufficient depth and breadth of information – gene or protein expression – by the identified cell populations). However, as described above in response to reviewer 3 point 4, in revision we have performed MAXFUSE to match the single cells resolved within our IMC analyses with cells in a publicly available scRNAseq dataset of *NF2* mutated VS tumours (Barret *et al* Nat Comms 2023: PMID: 38216553).

Consequently, in the revised manuscript we have performed targeted receptor-ligand interaction analyses between IMC and scRNA-seq aligned cell populations that we statistically proved were colocalised in different Antoni A and Antoni B regions of *NF2*-SWN VS tumours through cross pair correlation analysis (Figure 3). These analyses predict that T cells interact differently with myeloid cell populations and schwann cells within the VS tumours, with chemokine, cell adhesion and co-inhibitory pathways likely influencing differential T cell positioning and activities within the different tumour niches. We have added these new analyses as a new figure (**Figure 4**) and we have described and interpreted the results in the revised manuscript (**lines 280-315, 505-520**).

We believe these new data add substantial depth to the investigations in our manuscript and provide new insight into pathways that may be therapeutically targetable in *NF2* SWN patients.

3.12 Discussion on TAMs and Treatment Resistance: The manuscript should include further discussion on how the increased ratio of classically-activated to alternatively-activated TAMs contributes to treatment resistance or changes in the tumor microenvironment in bevacizumab-treated samples.

Response: We have added text into the discussion (**lines 448-452**) of the revised manuscript to address this comment.

Reviewer #5 (Remarks to the Author):

*In the manuscript entitled “Spatial mapping of immune cell environment in *NF2*-related Schwannomatosis vestibular Schwannoma”, Jones *et al.* applied Imaging mass cytometry to 13 cases of vestibular schwannoma with *NF2* mutations, to define the tumor microenvironment, revealing a different cell composition and cell-to-cell interaction in Antoni A vs Antoni B niches. The authors identified 23 different cell phenotypes, with several TAMs and Schwann cells phenotype described and being the major component of the TME in VS. The authors described several differences in cell-to-cell interaction in the Antoni A vs Antoni B areas. Additionally, several cellular neighborhoods have been identified and the reciprocal interaction have been assessed in Antoni A and B niches. Finally, the authors also compared TME of three (3) patients treated with Bevacizumab prior tumor resection and immune microenvironment of tumor treatment-naïve, identifying CD44+ Schwann cells has more abundant after Bevacizumab treatment, while AGTR1+ CD8+ TEMRA were reduced and potentially related to intratumoral remodeling. Vestibular Schwannoma with *NF2* mutation is a rare disease and as such its cellular landscape is not well characterize, however the manuscript focus has been placed on the cell-cell interaction analysis performed via cross pair correlation functions and cellular neighborhood analysis done by applying CellCharter and GMM.*

5.1 Strengths of this study include detailed characterization of a rare disease using spatial proteomics. This is also a limitation as the sample size is small (13 cases and 3 with prior

VEGF treatment) without validation using an external dataset. Further, the quality of the clustering is impacted by marker distributions that are not representative of immune subsets shown in Figure 1c, where there are many CD3⁺ clusters that do not express either CD4 or CD8. One CD3⁺ cluster is labeled as a TAM (light blue), while another is labeled as vascular endothelial cells (pink). CD163 is diffusely positive. This impacts all the downstream analysis and interpretation of spatial relationships. Together, with a low sample size and issues with staining and clustering, reduces the potential impact of this manuscript.

Response: We thank the reviewer for their detailed and constructive evaluation of our manuscript.

With respect to sample size, we have addressed this comment in response to reviewer 3 point 2.

We have also now provided extensive multi-modal validation of our analyses and cell identification using a publicly available sc-RNA-seq dataset of *NF2* mutated VS tumours, as described in our response to reviewer 3 point 4

We appreciate the reviewer's comment regarding the cell segmentation, clustering and identification within our study. We address some specific comments related to this in our responses below. However, it is now clear that double negative CD4⁻ CD8⁻ CD3⁺ T cells are found in a variety of tumours, and that these can play functionally important roles (Wu *et al* Front Immunol 2022). As such, the identification of these cells in our dataset does not necessarily represent failures in cell clustering and classification. We have added a sentence in the revised manuscript (**line 490-492**) to highlight this point.

The reviewer is correct that certain markers (such as CD163 and CD3) exhibit low level expression across different clustered cell populations. This is largely due to the nature of IMC analyses, where single cell segmentation and identification is done *in situ* within the context of the tissue environment, where cells are often densely packed and where they can have irregular morphologies (particularly within solid tumours). This can make it challenging to completely segregate marker expression between interacting cell populations.

In our analyses, we employed a conservative and extensively used cell segmentation approach to try to make cell identification as accurate as possible. This involved using a well-established pipeline for nuclear segmentation of IMC data (Windhager *et al* Nature Protocols 2023; Milosevic Bioinformatics Advances 2023), followed by a 1 pixel expansion to capture the cell cytoplasm (Maldegem *et al* Nat Comms 2021). As a one pixel expansion from nuclei (relating to 1µm within IMC staining) did not reach or pass the cell membrane of most cells, this did reduce the overall detection of marker expression within a cell, but it importantly limited capturing any molecules expressed on closely localised cells, and has been shown to increase the signal-to-noise ratio versus approaches which try to capture the entire cytoplasm (Maldegem *et al* Nat Comms 2021). In revision, we statistically demonstrate the success of this segmentation approach utilising Jaccard index analysis (which compares the manual vs automatic segmentation of the same cells), the gold standard approach for quantifying segmentation efficiency (Greenwald *et al* Nature Biotechnology 2021). This shows that the performance of our segmentation was good, and very similar to similar approaches based upon machine-learning driven nuclear segmentation (Greenwald *et al* Nature Biotechnology 2021)(**Fig S1A**). Notably, we tried using the REDSEA algorithm to counteract the bleed through of signal from neighbouring cells (Bai *et al* Front Immunol 2021), but found that it

also reduced the specific staining and did not give an overall increase in signal-to-noise in our IMC data.

We also undertook several validation steps before deciding on the final nomenclature of cell populations. Firstly, we examined whether related named cell populations resided within similar spaces in the Leiden clustering UMAP, with the rationale that different T cell sub populations should reside in the T cell region, myeloid cells should cluster within the myeloid cell region etc (as we showed in **Fig S1C, D**). We also evaluated the relative expression level of each marker on each clustered cell population and how this related to the expression of other canonical cell classification molecules, which also guided any required further sub-clustering of populations. We also confirmed segmented and clustered cell populations by backgating. We now show T cell and myeloid cell back gating relating to Iba1, CD163, CD3 and CD8 staining within **Reviewer Figure 1**.

Overall, whilst we believe we carefully and accurately segmented, clustered and named the cell populations used in our IMC study, low level “bleed through” of molecules from interacting cells is a feature of IMC work and we have now indicated this in our revised manuscript, along with our reasoning that our characterisation is accurate (**lines 144-152**). Importantly, in revision, our cell classification has also been orthogonally validated through multi-modal alignment with available VS scRNA-seq datasets, as described in response to reviewer 3 point 4, which further increases the confidence in our IMC clustering and analyses.

Additional points that need further clarification:

5.2 It seems that authors have used CD11b as marker for identification of neutrophils. However, CD11b is a marker expressed by several immune cells, including monocytes and macrophages, since it mediates immune cells migration and chemotaxis. Additionally, for this cluster there is no granzyme B expression. The only marker that can be used alone for neutrophil identification is CD66b, which is not included in this panel. Thus, it is hard to assign with high degree of certainty this phenotype based on the sole expression of CD11b. How do the authors justify the phenotype assignment? Have the authors tried other clustering algorithms which might provide a better clustering and improve the accuracy of cell phenotypes?

Response: We employed Leiden clustering for our IMC data, which is one of the most widely used and accepted approaches in the field. Leiden clustering is an improvement on the louvain community detection algorithm (Traag *et al* Scientific Reports 2019), which itself underlies other commonly used clustering algorithms used for IMC and CyTOF data such as PhenoGraph (Windhager *et al* Nature Protocols 2023, Weber and Robinson Cytometry part A 2016).

We acknowledge that we did not have CD66b in our panel to definitively identify neutrophils. From previous analyses, we do know that CD66b⁺ neutrophils express extremely high levels of CD11b (as shown in **Reviewer Figure 2**, demonstrating CD66b⁺CD11b⁺ neutrophils in the post-mortem lung of a deceased covid-19 infected adult). Moreover, the lack of expression of other myeloid cell markers (such as Iba1), precluded the annotation of CD11b very high cells as monocytes or macrophages.

To more convincingly confirm that CD11b⁺CD66b⁺ neutrophils are found in VS tumours, in revision we have performed two colour CD11b and CD66b IHC staining on VS tumours. We show that vascular associated CD66b⁺ cells are clearly observed in VS tumours, with CD66b being expressed on CD11b^{hi} cells. This new data is presented in **Fig S2** and described on **lines 120-122** in the revised manuscript.

5.3 How did the authors define alternatively activated and classically activated TAMs? Alternatively-activated TAMs are actually expressing higher level of PanCK, S100B compared to CD163 and CD68 and Iba1 (which I think was used as the most reliable marker for macrophage identification, but it still presents a lower expression compared to markers used for Schwann cell). CD68 expression is null in classically-activated TAMs. It is worth teasing out cells in these clusters in order to obtain cleaner clusters/phenotypes. Alternatively, the authors should support their choice in phenotype assignment.

Response: We defined alternatively activated TAMs (defined as alternatively activated-like TAMs in the revised manuscript) through the collective expression of Iba1, CD68, HLA-DR, Tim3 and CD163, with CD163 used to characterise the TAMs as alternatively activated-like. CD163 has been utilised in a number of other studies to classify macrophage subsets (Jayasingham *et al* Frontiers in Oncology 2020). We identified classically activated TAMs (now referred to as classically activated-like TAMs) through expression of CD74, lack / low expression of CD163 and high expression of Iba1. We acknowledge that this does not address the functionality of classically activated TAMs or provide in depth evaluation of other canonical classically activated markers, such as iNOS.

We agree that the segmented alternatively activated TAMs do appear to express PanCK and S100B, which we also use to distinguish schwann cell populations (**Figure 1C**); however, the Schwann cells do not express myeloid markers (e.g. iba1), allowing us to distinguish alternatively activated cells and PanCK expressing Schwann cells. As described above (response to reviewer 5 point 1), the overlap in expression of PanCK and S100B on identified alternatively activated macrophages, likely reflects the fact that alternatively activated-like macrophages and schwann cells are on occasion within the same cell neighbourhoods, particularly within Antoni A zones (as shown in **Figure 5A**), and that it can be challenging to segregate marker expression between adjacent and irregularly shaped cells, particularly when molecules (such as S100B) can also be deposited locally when the TME. However, it has previously been suggested within a scRNA-seq study that myeloid cells can express S100B within VS tumours (Xu *et al* AJP 2022). Moreover, we cannot discount the possibility that alternatively activated TAMs may express PanCK and S100B due to phagocytosing Schwann cells. As classically activated-like TAMs appear to localise infrequently with Schwann cells within the same cell neighbourhoods (as shown in **Figure 5A**), this may also provide explanation for the lower expression of PanCK and S100B on these cells. We provide text (**line 144-152**) within the revised manuscript to cover this point.

We attempted to sub cluster the alternatively activated-like TAM population to see if we could obtain cleaner and biologically meaningful sub populations, however this did not provide significantly clearer definition of CD163⁺S100B⁻, PANCK⁻ sub-populations. Moreover, it emphasised that macrophages exist on a spectrum of activation and phenotypic states in NF2-SWN VS tumours, as we state in more detail within the revised manuscript (**lines 137-143, 443-452**). As such, whilst it is possible to sub-cluster macrophage populations in greater and greater resolution, such an approach where distinct populations are sub-divided based upon very small biological differences will significantly undermine the power of the statistical analyses in the manuscript and also reduce the clarity of the observations in the manuscript.

Whilst we believe that we have clustered and identified an intuitive set of macrophage populations based upon the literature, we do agree that the signature and spectrum of cell states encompassed within the delineated alternatively activated and classically activated populations makes the definitive cell labelling misleading. As such, in revision we have reclassified the alternatively activated macrophage population as alternatively activated-like macrophages and

the classically activated macrophages as classically activated-like macrophages (**line 137-143** in revised manuscript). We do not believe that this alters the novelty or the conclusions in our manuscript but that it more accurately reflects the macrophage populations identified in our dataset, using the markers that we employed, and the cells within the NF2-SWN VS tumours.

5.4 How do the authors explain the higher expression of Tim3 in Schwann cells compare to T-cells or other immune cells? While the authors have 40 markers in their panel, some of those seem to be completely disregarded throughout the entire manuscript. This is especially true for signaling markers, which are not discussed or described in the context of the appropriate cell lineage. This is limiting the biological and immunological contribution this paper might provide.

Response: We apologize for this omission and the lack of description of certain markers in our results. This was due to the need to abridge our results section to align with word limits. In revision, we have expanded our description of our data to provide some insight into the expression of signalling molecules within clustered cell populations, and how they are used to inform cell activities (**lines 125-136**).

With respect to Tim3, it is known that this molecule can be widely expressed on a large number of cell types, across immune, stromal and neoplastic lineages (Dixon *et al* Science Immunology 2024, Alex *et al* ESMO Open 2019, Fang *et al* Cell, Death & Disease 2023). Indeed, although the reviewer is correct that Tim3 appears to be expressed by Schwann cells, it is also highly expressed by certain myeloid cell and T cell populations in our study (**Figure 1C**).

5.5 CD206 is used as marker for M2 polarized macrophage recognition, which has no expression in the present study. Since the authors preprocessed the raw images using IMC-denoise, do they altered the expression of this marker while cleaning the data? Usually, CD206 gives a good signal which helps in macrophages classification. Can the authors clarify this point?

Response: We thank the reviewer for highlighting this point. We agree that our Leiden clustering heat map in Figure 1C gives the impression that there is no CD206 expression by any cell population in our analyses and that the staining for this molecule, therefore, failed or was lost during denoising.

We have revisited our denoised images to assess CD206 expression. In the Leiden clustering UMAP, we can clearly see a small CD206 expressing sub population of cells within the population space (**Fig S1B** within the revised manuscript). Within images analysing CD206 expression, we can observe rare CD206⁺ Iba-1⁺CD163⁺ cells surrounding vessels, indicating that perivascular macrophages predominantly express CD206 in the NF2-SWN VS tumours (images provided in **Fig S2B** in the revised manuscript). Consequently, CD206 staining was successful in our panel, it is just not represented within the clustered cell populations in our analyses. This is due to the overall rarity of CD206 expression (approximately 0.003% of all myeloid cells), so that there was not sufficient weighting from this molecule (at our clustering resolution) to sub characterise CD206^{hi} populations of cells. In our revised manuscript we have added additional text to the results to address this point (**line 122-125**).

5.6 One of the major information missing is the number of cells per clusters and the total number of cells they have in the single cell database. Can the authors provide the percentage of each cluster they have identified in the dataset and for each patient? What is the number

or percentage of cell in “others” the unspecified cluster? How many cells were not classifiable based on the current panel?

Response: We agree with the reviewer that these are important information. In our revised manuscript we have added a table (**Table 3**) indicating the total number of cells in each cluster identified across our dataset and the relative percentage of each population across our dataset. We have also provided the relative percentage of the cell populations (out of total identified cells) in each case. We also provide information on the number (and percentages) of cells that were unclassifiable in our dataset and in each case.

5.7 Figure 1C: with such a large number of clusters, the authors should re-organize the heatmap such that the clusters order in the graph follows the order of clusters in the legend. This would make the interpretation and reading of the heatmap easier for the readers.

Response: We appreciate that the reading and the interpretation of the heat maps is difficult. We had attempted to group the legends so that the different sub-populations within the main groups could be tracked easily, but we recognise that this makes it difficult to align the populations within the differentially ordered heat map. In revision, we have modified the presentation of data within **Figure 1C**. We have also altered the presentation of the cell neighbourhood data (**Figure 5A**). We hope that these changes make the data easier to interpret.

5.8 How many Antoni A and B region were examined in the present study? How many per patient?

Response: We analysed a total of 73 ROIs from 13 patients (excluding bevacizumab treated cases in first part of analyses). This included 63 Antoni A ROIs, with 2-6 from each patient, and 10 Antoni B ROIs, with 1-4 from each patient. Although we analysed fewer Antoni B than Antoni A ROIs, due to the rarity of this region in many of the tumours, our dataset ensured that we had representation of Antoni B ROIs from multiple cases, to confirm robustness of our observations. We have now added this data into the revised manuscript (**line 102-104**).

5.9 Fig 1F: the study is examining 16 cases of VS, 13 treatment naïve and 3 post treatment. The first part of the paper seems to describe 13 naïve cases (line 97), however, in the figure 1F there are 16 cases listed. Later, they discussed the CN in naïve versus bevacizumab-treated cases. Can the authors clarify if the three treated patients were considered twice in the analysis (before and after resection)? It is confusing whether the first CN analysis and in general the first part of the paper is done on 13 cases excluding the three bevacizumab-treated cases or not, especially looking at Fig1 where there are 16 cases reported. The authors should clarify the case selection in the first part of the paper.

Response: We apologise for this oversight. Within our initial analyses we had included Bevacizumab cases from the start of the manuscript. However, the three Bevacizumab cases were subsequently removed from the early phases of the manuscript during restructuring. We have now removed these cases from **Figure 1F**, and to confirm, Bevacizumab cases were only used for analyses in **Figure 6**. We have also adjusted the case numbers in Table 1, to ensure cases are ordered logically for easier interpretation.

5.10 Fig4A: the legend and the dot plots are difficult to follow. Please re-organize the CNs based on their order in the legend, so it is easier to cross-check the information.

Response: We have modified the presentation of these data (Figure 5A in the revised manuscript).

5.11 Last part of the paper is confusing and should be revised. The authors are referencing supplemental figures not provided (there is no Fig 6B in the supplemental material). Line 325 is referring to the wrong figure (it should 5F instead of 5E).

Response: We apologize for this error, which has been corrected.

5.12 Supplementary fig 5E is describing something not reported in the actual figure. (there is no elbow plot).

Response: We have corrected this error in the revised manuscript.

5.13 The authors should mention the limitations of their study, which include at least the small sample size and the lack of important markers, such as PD1, especially in light of their finding and the emphasis they place in the presence of PDL1 Schwann cells, which have been described has one of the clusters with the greatest number of interactions. Thus, it would be important to have PD1 and mention this aspect as a limitation of the study.

Response: As discussed in response to reviewer 3 point 2, we have added text within the revised manuscript (**lines 439-441**) to state the limitation of small sample size, and the reasons for this within the study.

We acknowledge the reviewer's comment on the lack of PD-1 within our IMC antibody panel. We spent a substantial amount of time attempting to optimise different clones of anti-PD-1 antibodies for inclusion in our study; however, we were unable to obtain satisfactory staining. In the revised manuscript we indicate the limitation that PD-1 was not within the IMC panel but we also provide new qualitative IHC images (**Fig S6**) showing that CD8⁺ T cells within VS tumours express PD-1 (data also described on **line 251-253**) of the revised manuscript). We believe these data provide additional evidence that the PD-1-PD-L1 pathway is operational within VS tumours, potentially limiting CD8⁺ T cell activities.

5.14 Line 269: "biological processes outlined above and appeared to occupy distinct tissue regions as shown in the representative images in Fig 4C": which biological process are they referring to? This sounds unclear and the concept should be clarified as the figure 4C is a neighbor maps with several cell types.

Response: We apologize for the unclear statement. We have rewritten these sentences within the revised manuscript (**lines 346-347**).

5.15 Minor comments:

- 1) Abstract line 39-40: revise the sentence " Interestingly, T-cell populations associated with tumour-associated macrophages (TAMs) in Antoni A regions, seemingly limiting their ability to interact with tumorigenic Schwann cells". Did the authors meant ...are associated?
- 2) Line 53: "...the severity phenotype..." should be the severity of the phenotype?
- 3) Line 344: "greater proportion of the myeloid compartment"...the proportion has never been mentioned so provide this information is pivotal.

Response: We thank the reviewer for identifying these errors and omissions. These have been corrected in the revised manuscript.

Reviewer Figures

Reviewer Figure 1: Imaging mass cytometry analysis showing (A) nuclear identification and single cell segmentation within an Antoni A NF2-SWN VS tumour region of interest. (B) the expression level of CD3 and CD8 α T cell markers on clustered T cell and myeloid cells within NF2-SWN VS tumours; top row showing raw staining of CD3 and CD8; middle row and bottom row showing the calculated mean cell intensity of CD8 and CD3 across the identified cells, respectively. (C) the expression level of Iba1 and CD163 myeloid markers on clustered T cell and myeloid cells within NF2-SWN VS tumours; top row showing raw staining of Iba1 and CD163; middle row and bottom row showing the calculated mean cell intensity of Iba1 and CD163 across the identified cells, respectively (N/B closely associated cells are fused together in B, C due to nature of visualisation within Napari).

Reviewer Figure 2: Imaging mass cytometry analysis showing the identification of CD66b⁺CD11b^{hi} neutrophils in post-mortem lung of a patient with fatal covid-19. Full analyses in Da Silva Filho *et al* Sci Trans Med 2024

NCOMMS-24-28073A

We thank the editor and reviewers for their constructive and positive comments on our manuscript. We provide a point-by-point response to the reviewers' minor comments, below. All required textual changes within the revised manuscript are highlighted in **yellow**.

REVIEWER COMMENTS

Reviewer #2 (Remarks to the Author):

-Line 149: schwann cells should be capitalized

Response: This has been corrected.

-Line 252: "was not within the IMC, panel" – comma Is not in the correct location, should be "was not within the IMC panel,"

Response: This has been corrected.

-bevacizumab and losartan are generic names of drugs so should not need to be capitalized

Response: This has been corrected.

-Was CD Bev-0 significantly enriched in the bevacizumab treated samples? Based on the mean proportion graph in Figure 6E, there appears to be a greater difference than for the CN Bev-9, which was significantly different.

Response: We performed equivalent statistical analyses of the mean proportions of Bev-0 and Bev-9 cell neighbourhoods (normality test followed by Unpaired T test with Welch's correction) between the bevacizumab-treated and treatment-naïve cases. The p value = 0.020 for the relative proportion of the Bev-9 cell neighbourhood between the bevacizumab-treated and treatment-naïve samples. In contrast, the p value = 0.0838 for the relative proportion of the Bev-0 cell neighbourhood between the bevacizumab-treated and treatment-naïve samples. Although there was a noticeable difference in the mean proportion values for the Bev-0 cell neighbourhood between the groups, the higher standard deviation in the Bev-0 compared with Bev-9 cell neighbourhood analyses led to the above-threshold p value.

Reviewer #3 (Remarks to the Author):

Sample size limitation: While the authors provided a detailed explanation for the small sample size due to the rarity of NF2-related schwannomatosis, they could have more directly addressed how this limitation impacts the generalizability of their findings. A more explicit discussion of this limitation in the manuscript would be beneficial.

Response: We have added a sentence into the discussion of the revised manuscript (**lines 450-452**) to explicitly consider the generalizability of our results, given the small sample size in our study.

Antoni B regions in bevacizumab-treated samples: The authors explained the absence of

Antoni B regions in bevacizumab-treated samples, but they could have elaborated more on the potential biological significance of this observation in the manuscript itself.

Response: We have added a sentence into the discussion of the revised manuscript (**lines 562-564**) to question whether bevacizumab treatment influences the histopathological status of the tumour. We do not think it is appropriate to add more detailed interpretation on this point as we also indicate that future work will be required to confirm this hypothesis, due to the small number of cases evaluated.

Marker limitations: While the authors acknowledged the limitations of their 40-marker panel, they could have provided more discussion on how this might have impacted their ability to identify certain cell subsets, particularly in light of the reviewer's concerns about CD3+ clusters lacking CD4 or CD8 expression.

Response: We acknowledge the reviewer's point. In the previous revised version of the manuscript we specifically indicated that a limitation of the study was that the 40 antibody panel did not provide significant depth on the activation state and function of each cell type (**lines 441-445**). We have added to this in this revised version of the manuscript to indicate that it also made precise identification of cellular subsets challenging (**lines 445-446**).

Cell segmentation and clustering: The authors provided a detailed response regarding their segmentation and clustering approach, including validation steps. However, they could have incorporated more of this explanation into the manuscript itself to address potential concerns about the quality of cell identification.

Response: We added text in the results of the previous revised version of the manuscript to address this point (**lines 149-157**). In this revision version, we have also added text in the materials and methods to define our one-pixel expansion from the nucleus cell segmentation approach, and why this was selected for our project (**lines 656-658**).

Neutrophil identification: While the authors performed additional IHC staining to confirm CD66b+ neutrophils, they could have more explicitly addressed the limitations of using CD11b alone for neutrophil identification in their IMC panel within the manuscript.

Response: We have added text into the results of the revised manuscript to address this point and the importance of the IHC analysis to confirm that CD11b^{high} cells are neutrophils (**lines 123-125**).

TAM classification: The authors provided reasoning for their classification of alternatively activated-like and classically activated-like TAMs, but they could have incorporated more of this explanation into the manuscript to justify their phenotype assignments.

Response: We added substantial text into the previous revision version of the manuscript within the results and discussion to outline our rationale for the classification of the identified macrophage populations (**lines 141-157** and **lines 456-459**). We have, however, added additional text in the new revised manuscript to further address this point (**lines 144-145**).